Manuscript prepared for J. Name
with version 5.0 of the LaTeX class copernicus.cls.
Date: 5 April 2020

# Instrumental characteristics and potential Greenhouse gases measurement capabilities of the Compact High-spectral Resolution Infrared Spectrometer: CHRIS.

**Marie-Thérèse El Kattar**[1]**, Frédérique Auriol**[1]**, and Hervé Herbin**[1]

[1]Univ. Lille, CNRS, UMR 8518 - LOA - Laboratoire d'Optique Atmosphérique, F-59000 Lille, France

*Correspondence to:* Hervé Herbin
(herve.herbin@univ-lille.fr)

**Abstract.** Ground-based high spectral resolution infrared measurements are an efficient way to obtain accurate tropospheric abundances of different gaseous species and in particular GreenHouse Gases (GHG), such as $CO_2$ and $CH_4$. Many ground-based spectrometers are used in the NDACC and TCCON networks to validate the Level 2 satellite data, but their large dimensions and heavy mass makes them inadequate for field campaigns. To overcome these problems, the use of portable spectrometers was recently investigated. In this context, this paper deals with the CHRIS (Compact High-spectral Resolution Infrared Spectrometer) prototype with unique characteristics such as its high spectral resolution (0.135 $cm^{-1}$ non-apodized) and its wide spectral range (680 to 5200 $cm^{-1}$). Its main objective is the characterization of gases and aerosols in the thermal and shortwave infrared region, that's why it requires high radiometric precision and accuracy, which is achieved by performing spectral and radiometric calibrations that are described in this paper. Also, CHRIS's capabilities to retrieve $CO_2$ and $CH_4$ vertical profiles are presented through a complete information content analysis, a channel selection and an error budget estimation in the attempt to join the ongoing campaigns, such as MAGIC (Monitoring of Atmospheric composition and Greenhouse gases through multi-Instruments Campaigns), to monitor the GHG and validate the actual and future space missions such as IASI-NG and Microcarb.

## 1  Introduction

Remote sensing techniques have gained a lot of popularity in the past few decades due to the increasing need of a continuous monitoring of the atmosphere (Persky (1995)). Greenhouse gases, trace gases but also clouds and aerosols are detected and retrieved, thus improving our understanding of the chemistry, physics and dynamics of the atmosphere. Global scale observations are achieved using satellites, and one major technique is the InfraRed High Spectral Resolution spectroscopy (IRHSR). This technique offers radiometrically precise observations at high spectral resolution (Revercomb et al. (1988)), where quality measurements of absorption spectra are obtained. TANSO-FTS (Suto et al. (2006)), IASI (Clerbaux et al. (2007)) and AIRS (Aumann et al. (2003)) are examples of satellite sounders covering the Thermal InfraRed (TIR) region. The observations acquired from such satellites have many advantages such as: day and night data acquisition, possibility to measure concentrations of different gases, the ability to cover the land and sea surfaces (Herbin et al. (2013a)), and the added characteristic of being highly sensitive to various types of aerosol (Clarisse et al. (2010)). These spectrometers also have some disadvantages: local observations are challenging to achieve due to the pixel size that limits the spatial resolution, and the sensitivity in the low atmospheric layers is weak, where many short-lived gaseous species are emitted but rarely detected.

To fill these gaps, ground-based instruments are used as a complementary technique and one famous high-precision Fourier transform spectrometer is the IFS125HR from Bruker™, which will be briefly discussed in Section (3.5), further details can be found in Wunch et al. (2011). More than thirty instruments are currently deployed all over the world in two major international networks: TCCON (https://tccondata.org/) and NDACC (https://www.ndsc.ncep.noaa.gov/). This particular instrument has a very large size (1x1x3 m) and a mass well beyond a hundred kilograms, and therefore achieving a long optical path difference leading to a very high spectral resolution (0.02 and $5x10^{-3} cm^{-1}$ for TCCON and NDACC respectively). Despite its outstanding capabilities, this spectrometer is not suitable for field campaigns so it is mainly used to validate the Level 2 satellite data, thus limiting the scientifically important ground-based extension of atmospheric measurement around the world.

One alternative is the new IFS125M from Bruker (Pathatoki et al. (2019)) which is the mobile version of the well-established IFS125HR spectrometer. This spectrometer provides the highest resolution available for a commercial mobile FTIR spectrometer but it still has a length of about 2 m and requires on-site realignment by qualified personnel. Another alternative is the use of several compact medium-to-low resolution instruments that are currently under investigation such as a grating spectrometer (0.16 $cm^{-1}$), a fiber Fabry-Perot interferometer (both setups presented in Kobayashi et al. (2010)) and the IFS66 from Bruker (0.11 $cm^{-1}$) described in Petri et al. (2012). The EM27/SUN is the first instrument that offered a compact optically stable, high Signal-to-Noise Ratio (SNR), transportable spectrometer (Gisi et al. (2012)) that operates in the SWIR (Short Wavelength InfraRed) region. A new prototype called CHRIS (Compact High spectral Resolution Infrared Spectrometer) was conceived to satisfy some very specific characteristics: high spectral resolution (0.135 $cm^{-1}$, better than TANSO-FTS and the future IASI-NG); a large spectral band (680-5200 $cm^{-1}$) to cover the current and future infrared satellite spectral range and optimize the quantity of the measured species. Furthermore, this prototype is transportable and can operate for several hours on battery (>12H) so it is suitable for field campaigns. The full presentation of the characteristics and the calibration of this instrumental prototype is presented in Section 2.

Since carbon dioxide ($CO_2$) and methane ($CH_4$) are the two main greenhouse gases emitted by human activities, multiple campaigns have been worked out such as the MAGIC (Monitoring of Atmospheric composition and Greenhouse gases through multi-Instruments Campaigns) initiative, to better understand the vertical exchange of these GHG along the atmospheric column and to contribute to the preparation and validation of future space missions dedicated to GHG monitoring. CHRIS is part of this ongoing mission and this work presents for the first time the capabilities of such a setup in achieving GHG measurements,

where the forward model, state vector and error analysis is explained in Section 3.

In this context, we present in Section 3.5, a complete information content study for the retrieval of $CO_2$ and $CH_4$ of two other ground-based instruments that also participated in the MAGIC campaign: a comparison study with the IFS125HR instrument; and since CHRIS and EM27/SUN have a common band in the SWIR region, a study to investigate the spectral synergy in order to quantify the complementary aspects of the TIR/SWIR/NIR coupling for these two instruments. Moreover, Section 4 describes the channel selection made in this study. Finally, we summarize our results and perspectives for future applications; in particular, the retrieval of GHG in the MAGIC framework.

## 2  The CHRIS spectrometer

CHRIS is an instrumental prototype built by Bruker™ and used in different domains of atmospheric optics where its recorded spectra contains signatures of various atmospheric constituents such as GHG ($H_2O$, $CO_2$, $CH_4$), and trace gases. The capacity to measure these species from a technical point of view as well as the characterization of this prototype in terms of spectral and radiometric calibrations is presented in the following subsections.

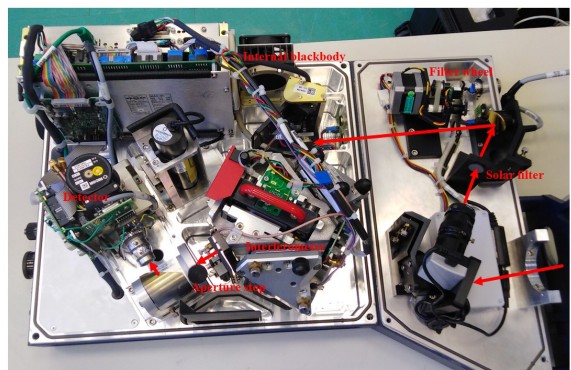

Fig. 1: An internal look of CHRIS, the red arrows illustrates the optical path of the solar beam inside the spectrometer.

### 2.1  General characteristics

CHRIS is a portable instrumental prototype with a mass of approximately 40 kgs and dimensions of 70x40x40 cm which makes it easy to operate on the field. The tracker, similar to the one installed on the EM27/SUN and described in details in Gisi et al. (2012), leads the solar radiation through multiple reflections on the mirrors to a wedged fused silica window.

An internal look of CHRIS is shown in Fig. 1, where the optical path of the solar beam is represented with red arrows: after multiple reflections on the tracker's mirrors, the solar radiation enters the spectrometer through the opening and is then reflected by the first mirror where the CCD camera verifies the collimation of the beam on the second mirror having a solar filter. At this level, CHRIS have a filter wheel that can be equipped with up to 5 optical filters with a diameter of 25mm. Filters are widely used when making solar measurements to reduce noise and non-linearity effects. After reflection on the second mirror, the beam enters the RockSolid™ Michelson interferometer which has two cube-corner mirrors to ensure the optical alignment stability of the beam and a KBr beamsplitter. After that, the radiation is blocked by an adjustable aperture stop which can be set between 1 and 18 mm. This limits the parallel beam parameter, and can be used to reduce the intensity of the incoming sunlight in case of the saturation of the detector. The remaining radiation falls on a MCT (Mercury Cadmium Telluride) detector, then it is digitized to obtain the solar absorption spectra in arbitrary units. This detector uses a closed-cycle stirling cooling system (a.k.a. cryo-cooler) so no liquid nitrogen has to be used. As the vibrations of the compressor may introduce noise in the spectra (see Sect. 2.6), a high scanning velocity (120 KHz) is needed.

A standard non-stabilized He-Ne laser controls the sampling of the interferogram. The condensation of the warm humid air on the beamsplitter due to its transportation between cold and warm environments, is the main reason of the use of a dessicant cartridge, so the spectrometer can operate in various environmental conditions. CHRIS has also an internal blackbody, which can be heated up to 353 K to make sure that there is no drift in the TIR region, and it also serves as an optical source to verify regularly the response of the detector.

## 2.2 Measurements and analysis

CHRIS's method of data acquisition is explained as follows: the interferograms are sampled and digitized by an Analogue-to-Digital Converter (ADC), and then numerically resampled at constant intervals of Optical Path Difference (OPD) by a He-Ne reference laser signal controlled by the aperture stop diameter. In order to determine a suitable compromise between the latter and to avoid the saturation of the signal, measurements must be done in a clear (no clouds or aerosols) and non-polluted (no gases with high chemical reactivity) atmosphere. For this purpose, a field campaign was carried out at the "El Observatorio Atmosférico de Izaña" (28.30°N, 16.48°W) in the Tenerife island. This particular observatory site is high in altitude (2374 m), away from pollution sites and has an IFS125HR listed in both the NDACC and TCCON networks. Saturation of the CHRIS's detector is reached at a value of 32000 ADC. The MCT detector is known for its high photometric accuracy but it also exhibits a non-linear response with regard to the energy flux in cases of high incident energy. This led us to choose an aperture stop of 5 mm, which is the best compromise between saturation and incoming energy flux.

Each spectrum corresponds to the solar transmission light in the total atmospheric column in a Field Of View (FOV) of 0.006 mrad. The spectral range spans the region from 680 to 5200 $cm^{-1}$ (1.9 to 14.7 µm) which corresponds to the Middle InfraRed region (MIR). The water vapor causes the saturation we see between the bands, thus the 0 signal. Therefore, we divided the spectrum into 4 distinctive spectral bands presented in Table 3: BT (thermal band: 680-1250 $cm^{-1}$), B1 (1800-2300 $cm^{-1}$), B2 (2400-3600 $cm^{-1}$) and B3 (3900-5200 $cm^{-1}$) and this annotation is used for the rest of the paper.

## 2.3 Optical features

A technical study was conducted on this prototype in order to evaluate its optical and technical properties with a constant aperture stop diameter of 5 mm. One of the most important findings is the effect of the number of scan on the measured spectra. In practice, a scan is the acquisition of a single interferogram when the mobile mirror of the Michelson interferometer begins data collection at the Zero Path Difference (ZPD) and finishes at the maximum length, therefore achieving the highest resolution required. In Fig.2, the spectrum with 10 scans has a higher amplitude than that of 50 and 100 scans. On the other hand, the spectra of 50 and 100 scans are clearly less noisy than that of the 10 scans. This is due to the fact that the increase in the number of scans causes an increase in the SNR which leads to the decrease of the noise. However, there is a limit to the number of scans beyond which no improvement of the SNR is obtained. The SNR is proportional to the square root of the acquisition time (number of scans), also known as the Fellgett's advantage and since the detector is shot noise dominated, the improvement of the SNR with the number of scans is blocked at a certain value, and this is why the spectra of 100 and 200 scans do not show a significant difference. The SNR is an estimation of the root mean squared noise of the covered spectral domain and can be calculated in OPUS (the running program for CHRIS) using the function SNR with option fit parabola; it is estimated to be approximately 780. An optimized criteria is chosen to select the appropriate number of scans: when the wanted species has a fast changing concentration, such as volcanic plumes, a relatively small number of scans is needed to be able to follow temporally the change in the atmospheric composition; while in contrast, when measurements of relatively stable atmospheric composition are made, for example GHG ($CO_2$ and $CH_4$), the number of scans can be increased to 100. For instance, the time needed for 1 scan with a scanning velocity of 120 KHz is 0.83s, so 100 scans take approximately 83s which is low in comparison with the variability of $CO_2$ and $CH_4$ in the atmosphere.

Another important feature is the effect of the gain amplitude (and pre-amplifiers) on the spectra, which plays the role of amplifying the signal before digitization. Those parameters should be chosen in a way that the numeric count, falls in a region where no detector saturation occurs. If the gain is increased of a certain amount, the background noise is increased by the same amount. The use of such an option in the measurement procedure might be considered in cases where the signal is very weak, like the lunar measurements. Note that there are other ways to increase the intensity of the signal like using signal amplifying filters (see Section 2.1), or increasing the aperture stop diameter.

## 2.4 Radiometric and spectral calibration

In the following section, the spectral and radiometric calibrations are discussed in order to convert spectra from numeric counts expressed in arbitrary units to radiance expressed in $W.m^{-2}.sr^{-1}.cm$.

### 2.4.1 Radiometric calibration

Despite the fact that CHRIS has an internal blackbody, radiometric calibration cannot be overlooked because of its narrow spectral coverage (only the TIR region), and since the radiometric noise, time dependent and wavelength dependent calibration errors are magnified in the inversion process, high radiometric precision is required to derive atmospheric parameters from a

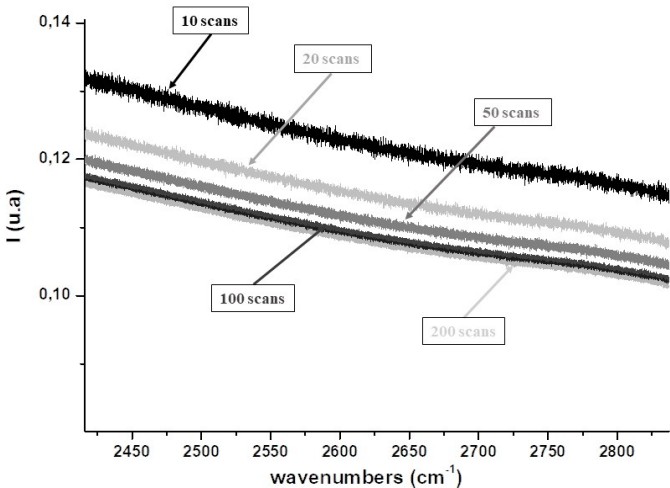

Fig. 2: Improvement of the SNR with different scan numbers and a fixed scan speed of 120 KHz.

spectrum. We calibrate our spectra using the two points calibration method explained in Revercomb et al. (1988). This method consists of using the observations of hot and cold blackbody reference sources which will be used as the basis for the two points calibration at each wavenumber. A cavity blackbody was acquired by the LOA to perform regular radiometric calibrations. The latter is an HGH/RCN1250N2 certified by the LNE (Laboratoire National de métrologie et d'Essais) as having an emissivity greater than 0.99 in the spectral domain spanned by CHRIS, a stability of 0.1 K at 1173 K, an opening diameter up to 50 mm (corresponding to that of CHRIS), and covering temperatures from 323 to 1523 K. This cavity blackbody is mounted on an optical bench and used before and after each campaign to perform absolute radiometric calibrations through open path measurements and make sure that this calibration is stable across the whole spectral range. These two blackbody temperatures are viewed to determine the slope $m$ and offset $b$ (Eq. 1 & 2) which defines the linear instrument response at each wavenumber. The slope and the offset can be written following Revercomb et al. (1988):

$$m = \frac{S_c - S_h}{B_\nu(T_c) - B_\nu(T_h)} \tag{1}$$

$$b = \frac{S_c * B_\nu(T_h) - S_h * B_\nu(T_c)}{B_\nu(T_h) - B_\nu(T_c)} \tag{2}$$

where $S$ is the blackbody spectrum recorded and $B_\nu$ corresponds to the calculated Planck blackbody radiance; the subscripts $h$ and $c$ corresponds respectively to the hot (1473 K) and cold (1273 K) blackbody temperatures. Finally, the calibrated spectrum expressed in $W.m^{-2}.sr^{-1}.cm$ is obtained by applying the following formula:

$$L = \frac{S - b}{m} \tag{3}$$

where S is the spectrum recorded by CHRIS.

### 2.4.2 Instrumental Line Shape (ILS) and spectral calibration

One open path measurement using the calibrated HGH blackbody as source was performed, similar to the one previously described in Wiacek et al. (2007), to record a spectrum without applying any apodization. Our colleagues in the PC2A laboratory provided us with a 10-centimeter-long cell, with a free diameter of 5 cm where the pressure inside is monitored by a capacitive gauge. With the help of the line-by-line radiative transfer algorithm, ARAHMIS (Atmospheric Radiation Algorithm for High-spectral Measurements from Infrared Spectrometer), developed at the LOA laboratory an MOPD of 4.42 cm was determined

corresponding to a spectral resolution of 0.135 $cm^{-1}$ using a sinc function with a spectral sampling every 0.06025 $cm^{-1}$ to satisfy the Nyquist criterion. In FTIR spectroscopy, a poor ILS determination generates a significant error in the retrieval process, so we are currently modifying the optical bench in order to perform an ILS determination at the same time as the radiometric and spectral calibrations before each field campaign.

The sampling of the interferogram is controlled by a standard, not frequency-stabilized He-Ne laser with a wavelength of 632.8 nm which serves as a reference while converting from the distance scale to the wavenumber scale. The instrument is subjected to changes in pressure and temperature since it operates in different locations, therefore in different meteorological conditions. This will cause a change in the refractive index and as a consequence a change in the reference wavelength of the laser which will lead to an instability in the conversion process and therefore the need for a spectral calibration to reduce this error. The ILS line defined above is used to re-simulate isolated absorption lines from the HITRAN database (Gordon et al. (2017)) considering non-apodized spectra which allows the exploitation of the full spectral resolution. In short, we choose an intense, non-saturated and always present in the spectra absorption lines of $H_2O$ with wavenumber $\nu$, then we compare it with the simulation and calculate its new wavenumber $\nu^*$:

$$\nu^* = \nu(1+\alpha) \tag{4}$$

where $\alpha$ is the calibration factor. Eq.4 is limited by a 0.038 $cm^{-1}$ precision, corresponding to roughly half of the spectral sampling, estimated from the standard deviation between the theoretical HITRAN spectroscopic lines and the measured ones by CHRIS. Fig. 3 shows the comparison between a calibrated and a non-calibrated spectrum along with the solar Planck function explained in Section 3.1. Spectral and radiometric calibrations procedure is automated using a Matlab code to convert the spectra instantly from numeric counts to absolute radiance.

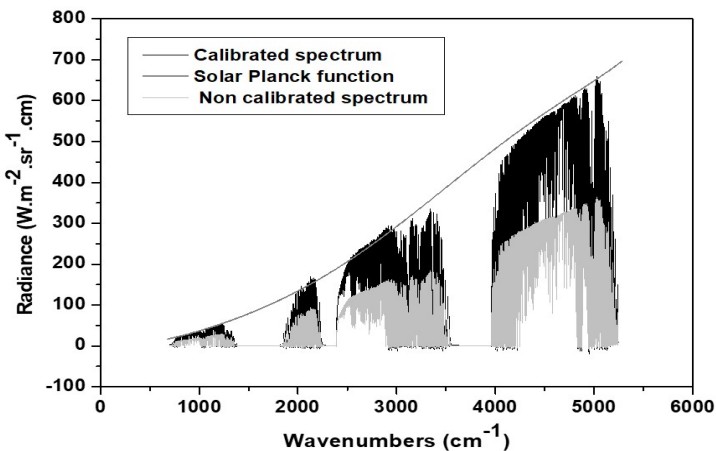

Fig. 3: The calibration process transforms the non-calibrated spectrum (light gray) into a calibrated one (black) that fits with the solar Planck function (solid gray line).

## 2.5 Radiometric stability

During campaigns and after long transportations, constant measurements of the internal blackbody, which can be heated up to 353 K, is carried out in the thermal infrared region (most affected by drifts). Figure 4 shows the variations of the internal blackbody during multiple field campaigns: a little fluctuation in function of the measurement conditions can be seen, but depending on the locations and even years, no systematic drift can be detected, so we can safely say that the instrument is quite stable between each laboratory calibration.

## 2.6 Spectral artifacts

There are commonly several well-known spectral artifacts: the aliasing, the picket-fence effect also known as the resolution bias error, and the phase correction, which are well controlled in CHRIS.

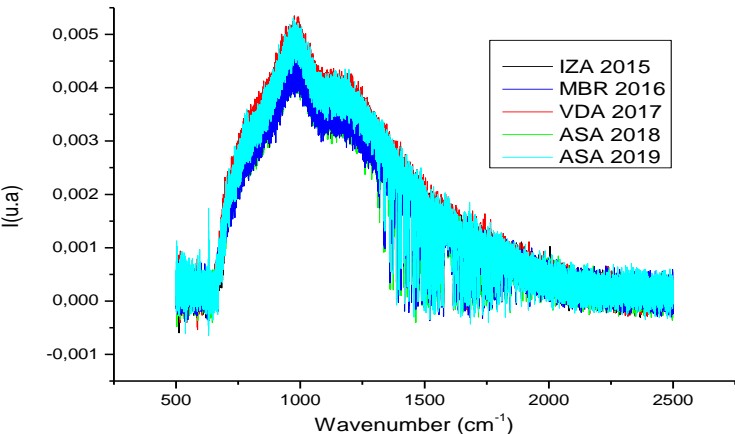

Fig. 4: The radiometric stability of the instrument is achieved by following the variations of the internal blackbody during multiple field campaigns. IZA : Izana, MBR : M'Bour, VDA : Villeneuve d'Ascq, ASA : Aire-sur-l'Adour.

An aliasing is the result of the missampling of the interferogram in the x-axis locations, which leads to errors in the retrieved column abundances due to its overlap with the original spectrum. The He-Ne laser, having a wavelength $\lambda$ of 632.8 nm, generates the sampling positions of the interferogram at each zero crossing. No overlap will occur if the signal of the spectrum is 0 above a maximum wavenumber $\nu_{max}$ and if $\nu_{max}$ is smaller than the folding wavenumber $\nu_f = 1/(2.\Delta x)$. Since $\Delta\nu$ is related to the sample spacing $\Delta x$, the minimum possible $\Delta x$ is 1/31600 cm, since each zero crossing occurs every $\lambda/2$. This corresponds to a folding wavenumber of 15800 $cm^{-1}$, i.e. the maximum bandwidth that can be measured without overlap has a width of 15800 $cm^{-1}$. This source error is of special relevance to the spectra acquired in the Near Infrared region. However, for the MIR, the investigated bandwidth is much smaller than 15800 $cm^{-1}$, where $\nu_{max}$ is less than 5200 $cm^{-1}$ so CHRIS's spectra are not affected by this problem (Dohe et al. (2013)).

The picket-fence effect, or the resolution bias error, becomes evident when the interferogram contains frequencies, which do not coincide with the frequency sample points, but this is overcome in our spectra by the classical method of the Zero Filling Factor (ZFF) where zeros are added to the end of the interferogram before Fourier Tranform is performed, thereby doubling the size of the original interferogram.

The phase correction is necessary while converting the interferogram into a spectrum, which is relevant to single-sided measurements, similar to those acquired by CHRIS. Mertz phase correction is the method used for CHRIS to overcome this problem, which relies on extracting the real part of the spectrum from the complex output by multiplication of the latter by the inverse of the phase exponential, therefore eliminating the complex part of the spectrum generated.

Besides these classical FTIR artefacts, we noticed during our tests that when using a scan speed of 160 KHz, we drastically increase the non-linearity effect of the detector (see Fig. 5). However, we identified a ghost signal for low scanning velocities (for example 40 KHz shown in Fig. 6). This ghost is specific to CHRIS because it is caused by the noise introduced from the vibrations of the compressor used in the closed-cycle Stirling cooler as mentioned in Sect. 2.1. The choice of a scanning velocity of 120 KHz is a compromise between two important features: the elimination of the ghost signal, which appears at scanner velocities below 80 KHz, and the increase of the detector non-linearity at a velocity of 160 KHz.

## 3  Information content analysis

Since CHRIS is an instrumental prototype, its ability to retrieve GHG is unknown; therefore it is important to perform an information content study to quantify, in a first attempt, its potential capability to retrieve GHG. In this context, CHRIS is one of the instruments deployed in the MAGIC project alongside satellites, lidar, balloons and ground-based measurements. The latter is a French initiative supported by the CNES (Centre National d'Etudes Spatiales), which aims to implement and organize

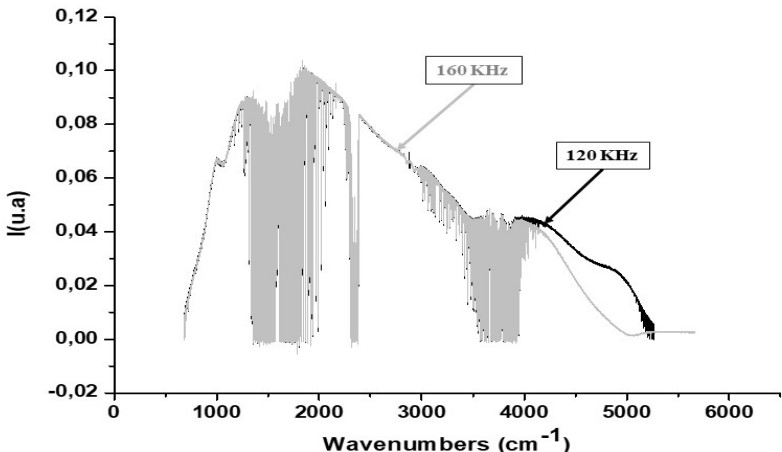

Fig. 5: Spectra of the external blackbody with two different scanning velocities: 120 KHz (black) and 160 KHz (light gray).

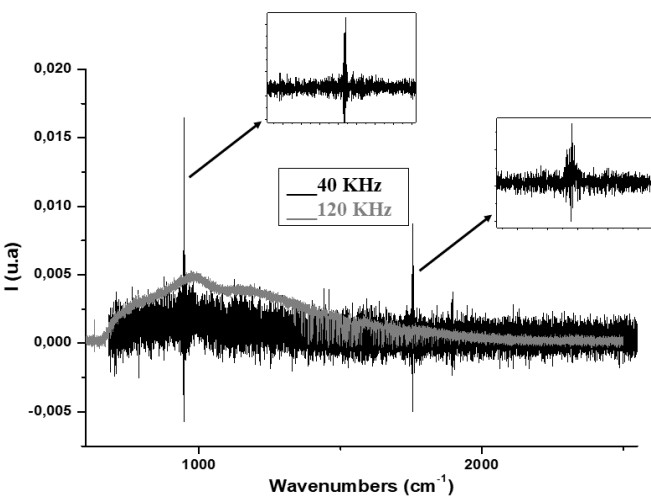

Fig. 6: Spectra of the internal blackbody of CHRIS with scanner velocities of 40 KHz (black) and 120 KHz (light gray).

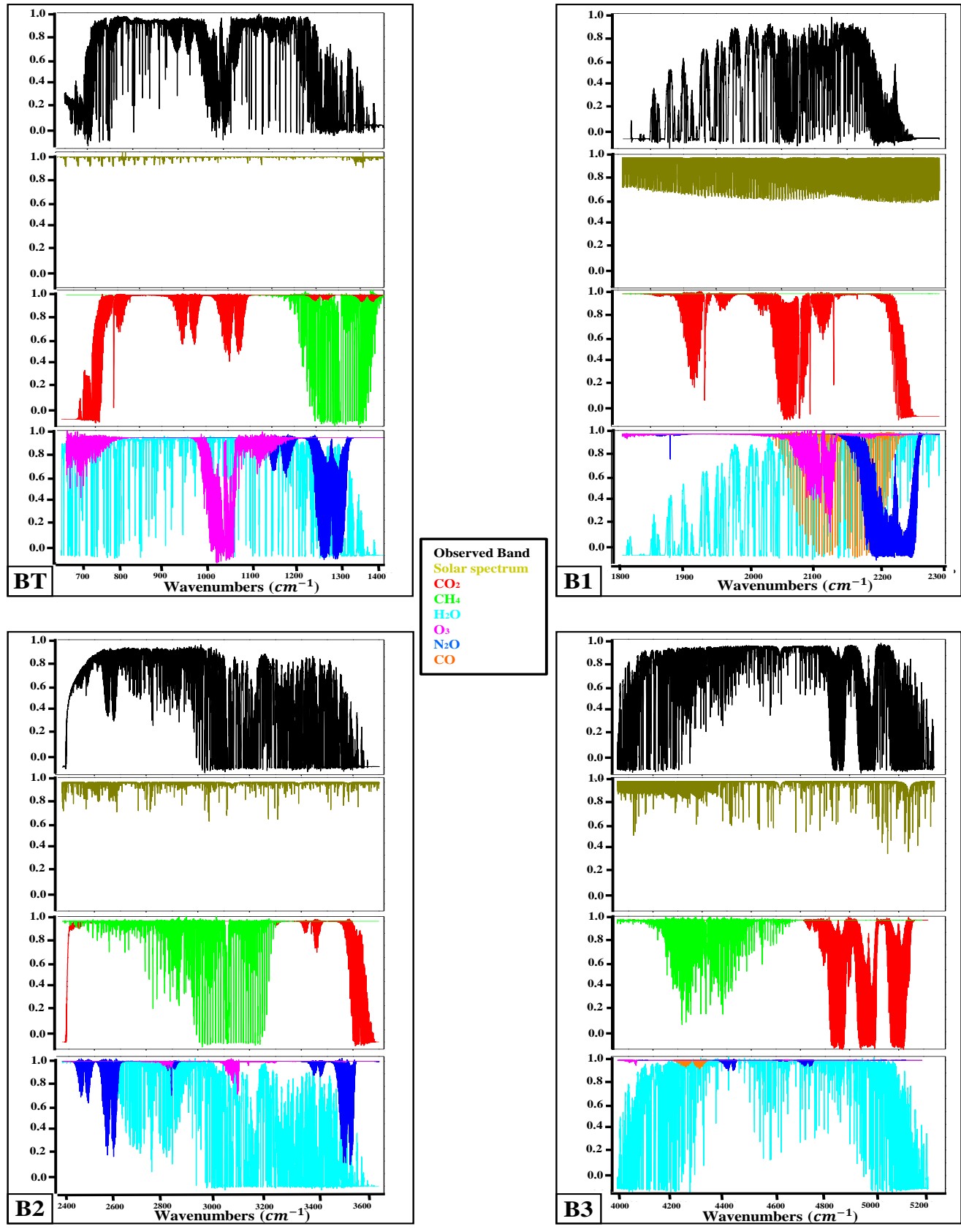

Fig. 7: Measured and simulated CHRIS spectrum in transmittance for clear sky conditions at Izaña observatory. Each band is calculated from the line-by-line forward model ARAHMIS and the solar pseudo-transmittance spectra reported by Toon (2015).

regular annual campaigns in order to better understand the vertical exchange of the GHG ($CO_2$ and $CH_4$) along the atmospheric column and establish a long-term validation plan for the satellite level 2 products.

## 3.1 The forward model

Accurate calculations of the radiances observed by CHRIS are achieved with the line-by-line radiative transfer algorithm ARAHMIS over the thermal and shortwave infrared spectral range (1.9-14.7 μm). Gaseous absorption is calculated based on the updated HITRAN 2016 database (Gordon et al. (2017)). The absorption lines are computed assuming a Sinc line shape and no-apodization is applied which allows the exploitation of the full spectral resolution. In this study, the term "all bands" refers to the use of the bands BT, B1, B2 and B3 simultaneously of CHRIS (see Section 2.2). Absorption continua for $H_2O$ and $CO_2$ are also included from the MT-CKD model (Clough et al. (2005)). The pseudo-transmittance spectra corresponding to direct sunlight from the center of the solar disk reported by Toon (2015) is used as the incident solar spectrum interpolated on the spectral grid of CHRIS. The effective brightness temperature depends strongly on the wavenumber, thus the Planck function is calculated in each spectral domain of CHRIS determined from Trishchenko (2006) who combined the work of four recent solar reference spectra. Two of these reference spectra with 0.1% relative difference are taken into consideration, and then adjusted by a polynomial fit (solid line in Fig. 3). In the gaseous retrieval process, the spectrometer's line-of-sight (LOS) has to be known for calculating the spectral absorption of the solar radiation while passing through the atmosphere. For this, the time and the duration of each measurement is saved, from which the required effective solar elevation (respectively the Solar Zenith Angle (SZA)) is calculated based on the routine explained in Michalsky (1988).

As mentioned in Section 3.3.2, $Iza\tilde{n}a$ offers clear non-polluted measurements since it is high in altitude and away from major pollution sites, so calculations are performed based on the concentration of the desired atmospheric profile with the corresponding profile information: the temperature, pressure and relative humidity are derived from the radiosondes (http://weather.uwyo.edu/upperair/sounding.html); $CO_2$ and $CH_4$ profiles are derived from the TCCON database, whereas $O_3$, $N_2O$ and $CO$ concentrations are calculated from a typical mid-latitude summer profile. Figure 7 shows the results of the forward model simulation superimposed with the four infrared bands measured by CHRIS. For each band, we present the influence of the solar spectrum, the GHG ($CO_2$ and $CH_4$) and the major interfering molecular absorbers. We can see the good agreement between ARAHMIS's simulations and the CHRIS measurements in clear sky conditions.

## 3.2 Information Content (IC) theoretical basis

Once the forward model is calculated, we rely on the formalism of Rodgers (2000) that introduces the optimal estimation theory used for the retrieval, which is widely described elsewhere (e.g. Herbin et al. (2013a)) and summarized hereafter.

In the case of an atmosphere divided in discrete layers, the forward radiative transfer equation gives an analytical relationship between the set of observations $y$ (in this case, the radiance) and the vector of true atmospheric parameters $x$ (i.e. the variables to be retrieved: vertical concentration profiles of $CO_2$ and/or $CH_4$):

$$y = F(x;b) + \varepsilon, \tag{5}$$

where $F$ is the forward radiative transfer function (here the ARAHMIS code), $b$ represents the fixed parameters affecting the measurement (e.g. atmospheric temperature, interfering species, viewing angle, etc.) and $\varepsilon$ is the measurement error vector.

In the following information content study, two matrices fully characterize the information provided by CHRIS: the averaging kernel $A$ and the total error covariance $S_x$.

The averaging kernel matrix A, gives a measure of the sensitivity of the retrieved state to the true state, and is defined by:

$$A = \partial \hat{x}/\partial x = GK, \tag{6}$$

where $K$ is the Jacobian matrix (also known as the weighting function), where the $i$th row is the partial derivatives of the $i$th measurement with respect to each ($j$) element of the state vector: $K_{ij} = (\partial F_i/\partial x_j)$, and $K^T$ is its transpose.

The gain matrix $G$, whose rows are the derivatives of the retrieved state with respect to the spectral points, is defined by:

$$G = \partial \hat{x}/\partial y = (K^T S_\varepsilon^{-1} K + S_a^{-1})^{-1} K^T S_\varepsilon^{-1}, \tag{7}$$

where $S_a$ is the a priori covariance matrix describing our knowledge of the state space prior to the measurement and $S_\epsilon$ represents the forward model and the measured signal error covariance matrix.

At a given level, the peak of the averaging kernel row gives the altitude of maximum sensitivity, whereas its full width at half maximum (FWHM) is an estimate of the vertical resolution. The total Degrees Of Freedom for Signal (DOFS) is the trace of $A$,

which indicates the amount of independent pieces of information that one can extract from the observations with respect to the state vector. A perfect retrieval resulting from an ideal inverse method would lead to an averaging kernel matrix $A$ equal to the identity matrix with a DOFS equal to the size of the state vector. Therefore, each parameter we want to retrieve is attached to the partial degree of freedom represented by each diagonal element of $A$.

The second important matrix in the IC study is the error covariance matrix $S_x$, which describes our knowledge of the state space posterior to the measurement. Rodgers (2000) demonstrated that this matrix can be written as:

$$S_x = S_{smoothing} + S_{meas.} + S_{fwd.mod.} \tag{8}$$

From Eq.8, the smoothing error covariance matrix $S_{smoothing}$ represents the vertical sensitivity of the measurements to the retrieved profile:

$$S_{smoothing} = (A - I)S_a(A - I)^T \tag{9}$$

$S_{meas.}$ gives the contribution of the measurement error covariance matrix through $S_m$ which illustrates the measured signal error covariance matrix, to the posterior error covariance matrix $S_x$. $S_m$ is computed from the spectral noise:

$$S_{meas.} = GS_mG^T \tag{10}$$

At last, $S_{fwd.mod.}$ and gives the contribution of the posterior error covariance matrix through $S_f$ the forward model error covariance matrix, which illustrates the imperfect knowledge of the non-retrieved model parameters:

$$S_{fwd.mod.} = GK_bS_b(GK_b)^T = GS_fG^T \tag{11}$$

with $S_b$ representing the error covariance matrix of the non-retrieved parameters.

### 3.3 A priori information

The IC analysis uses simulated radiance spectra of CHRIS in the bands: BT, B1, B2 and B3. The $CO_2$ and $CH_4$ vertical concentrations of the a priori state vector $x_a$ are based on a profile that follows the criteria described in Section 3.1 discretized by 40 vertical layers, extending from the ground to 40 km height with 1 km step. In addition, the water vapor vertical profile, the temperature and the SZA are included in the non-retrieved parameters and are discussed in Section 3.3.3. The a priori values and their variabilities are summarized in Table 1 and are described in the following sections.

### 3.3.1 A priori error covariance matrix

In situ data or climatology can give us an evaluation of the a priori error covariance matrix $S_a$. Since the use of diagonal a priori covariance matrices is common for the retrievals from space measurements (e.g. De Wachter et al. (2017)), and since this study is dedicated to information coming from the measurement rather than climatology's or in situ observations, we assume firstly that $S_a$ is a diagonal matrix with the $i$th diagonal element $(S_{a,ii})$ defined as:

$$S_{a,ii} = \sigma_{a,i}^2 \quad \text{with} \quad \sigma_{a,i} = x_{a,i} \cdot \frac{p_{error}}{100} \tag{12}$$

where $\sigma_{a,i}$ stands for the standard deviation in the Gaussian statistics formalism. The subscript $i$ represents the $i$th parameter of the state vector. The $CO_2$ profile a priori error is estimated from Schmidt and Khedim (1991). The $CH_4$ a priori error is fixed to $p_{error}$=5% similar to the one used in Razavi et al. (2009) for the retrieval of the methane obtained from IASI and also to be consistent with the previous study concerning the TANSO-FTS instrument (Herbin et al. (2013a)).

Nevertheless, the correlation of the vertical layers is more expressed by the off-diagonal matrix elements. This is the reason we also use an a priori covariance matrix similar to the one used in Eguchi et al. (2010), where the climatology derived from TCCON is used to construct this matrix. The study with these two covariance matrices is presented for CHRIS in the following sections.

### 3.3.2 Measurement error covariance matrix

The measurement error covariance matrix is computed knowing the instrument performance and accuracy. The latter is related to the radiometric noise expressed by the SNR already discussed in Section 2.3. This error covariance matrix is assumed to be diagonal, and the $i$th diagonal element can be computed as follows:

$$S_{m,ii} = \sigma_{m,i}^2 \quad \text{with} \quad \sigma_{m,i} = \frac{y_i}{SNR}, \tag{13}$$

where $\sigma_{m,i}$ is the standard deviation of the $i$th measurement $(y_i)$ of the measurement vector $y$, representing the noise equivalent spectral radiance. The SNR for CHRIS is estimated to be 780 and it is reported with other instrumental characteristics in Table 3.

| State vector elements | T | $H_2O$ | SZA | $CO_2$ | $CH_4$ |
|---|---|---|---|---|---|
| a priori values | Radiosondes | Radiosondes | 10°/80° | TCCON database | TCCON database |
| A priori uncertainty ($P_{error}$) | 1 K/layer | 10% | 0.35° | 1.3-8% | 5% |

Table 1: State vector parameters.

### 3.3.3 Non-retrieved parameter characterization and accuracy

The effects of non-retrieved parameters is a complicated part of an error description model. In our case these uncertainties are
limited to the interfering water vapor molecules due to its important existence in the spectra and the effect of the temperature,
where a vertically uniform uncertainty is assumed in both cases. It is important to note that in this study the water vapor is
considered as a non-retrieved parameter, for the sake of comparison with Herbin et al. (2013a), but it will be part of the retrieved
state vector in the inversion process which will be the subject of a future study.

On one hand, we assumed a partial column with an uncertainty ($p_{Cmol}$) of 10% instead of a profile error for $H_2O$. On the other
hand, we assumed a realistic uncertainty of $\delta T = 1K$, compatible with the typical values used for the ECMWF assimilation, on
each layer of the temperature profile. Moreover, we assumed a realistic uncertainty of 0.35° on the SZA, corresponding to the
difference in the solar angle during the acquisition of a measurement corresponding to 100 scans. All these variabilities are
reported in Table 1.

The total forward model error covariance matrix ($S_f$), assumed diagonal in the present study, is given by summing the
contributions of each diagonal element, and the $i$th diagonal element ($S_{f,ii}$) is given by:

$$S_{f,ii} = \sum_{j=1}^{nlevel} \sigma_{f,T_j,i}^2 + \sigma_{f,H_2O,i}^2 + \sigma_{f,SZA,i}^2 \tag{14}$$

Here, the spectroscopic effects such as the line parameter, the line mixing and the continua errors are not considered but they are
discussed with the $X_G$ column estimation in Section 3.4.2.

### 3.4 Information content analysis applied to greenhouse gases profiles

An information content analysis is performed on the whole spectrum for $CO_2$ and $CH_4$ separately to quantify the benefit of
the multispectral synergy. Separately means that the state vector is constituted of only one of the above gas concentrations at
each level between 0 and 40 km to match the altitudes reached by TCCON and the MAGIC instruments (balloons and planes
reaching altitudes of more than 25 km). This corresponds to the case where we estimated each gas profile alone when all other
atmospheric parameters and all other gas profiles are known from ancillary data with a specific variability or uncertainty. Two
different SZA (10° and 80°) are chosen to demonstrate the effect of the solar optical path on the study since the sensitivity is
correlated to the viewing geometry. Furthermore two different a priori covariance matrices are used to show the effect of using
climatological data describing the variability of GHG profiles. In the following subsections, we explain in details the averaging
kernel, error budget and total column estimations.

### 3.4.1 Averaging kernel and error budget estimation

Figure 8 shows the averaging kernel $A$ and total posterior error $S_x$ for $CO_2$ for an angle of 10°. The figures of the second SZA
(80°) are not shown since the vertical distribution of the kernels and errors is quite similar and exhibits only slight differences
in the amplitude with respect to the other angle. However, the results are different, this is why they are discussed in order to
quantify the information variability with the viewing geometry. $A$ is obtained for $CO_2$ independently using the variability
introduced in Section 3.3.1, and considering an observing system composed of the band BT, B1, B2 or B3 separately and all the
bands together to quantify the contribution of each of the spectral bands and show the benefits of the TIR/SWIR spectral synergy.
Each colored line represents the row of $A$ at each vertical grid layer. Each peak of $A$ represents the partial degree of freedom of
the gas at each level that indicates the proportion of the information provided by the measurement. In fact, if the value is close
to unity, it means that the information comes predominantly from the measurement, but a value close to zero means that the
information comes mainly from our prior knowledge of the a priori state. We can clearly see that at lower altitudes and up to 10
km, the kernels are close to unity suggesting that the measurement improved our knowledge, while at higher altitudes (beyond

10 km) the kernels are close to 0. It is important to also note that when using all the bands simultaneously, the information distribution of the kernels is improved and is more homogeneous along the vertical profile.

The measurement may provide information about $CO_2$ from the ground up to 20 km high in the atmosphere (all bands), while at much higher altitudes the information comes mainly from the a priori, due to a smaller sensitivity of these gases in the upper troposphere. This is clearly represented in the error budget study: the a posteriori total error (solid black line) is significantly smaller than the a priori error (red line) in the lower part of the atmosphere (between 0 and 20 km), which means that the measurement improved our knowledge of the $CO_2$ profile; while beyond 20 km, the total a posteriori error is equal to the a priori error suggesting a very poor sensitivity at high altitudes. Furthermore, one can notice that the measurement error stays very weak regardless of the band used which proves that the error related to the SNR is negligible. Also, the forward model error depending on the non-retrieved parameters remains quite modest. However, the smoothing error predominates the other errors and becomes preponderant beyond 20 km, which means that the information is strongly constrained by the a priori profile at high altitudes, and little information is introduced from the measurement. To overcome this problem, another similar study was conducted but with a non-diagonal a priori covariance matrix (Eguchi et al. (2010)). The vertical distribution is more homogeneous through all the layers. The shape of the error budget is very similar to that of the variance; however, the a priori and a posteriori errors are significantly reduced. The measurement and forward model errors remain weak, but it is important to note that despite the fact that the smoothing error is smaller, the constraint is stronger. This has the effect of decreasing the uncertainty, but increasing the propagation of the smoothing error along the vertical layers, which explains the smaller values of the DOFS.

Finally, the total DOFS for $CO_2$ are shown in Table 4 for angles of 10° and 80°. It shows that, for a diagonal a priori covariance matrix, one might be able to retrieve between two and three partial troposheric columns for $CO_2$, and as expected the DOFS is slightly higher at 80° since the optical path of the sun in every layer is longer. However, when using a non-diagonal a priori covariance matrix, one less partial tropospheric column is retrieved but with significant improvement in the error budget estimation.

The same reasoning is followed for $CH_4$: A is obtained for $CH_4$ independently using the variability introduced in Section 3.3.1, and considering an observing system composed of the band BT, B1, B2 or B3 separately and all the bands together. Fig.9 shows that the vertical distribution of $CH_4$ is more homogeneous than that of $CO_2$ and we can see that the $A$'s are broader than those of $CO_2$, suggesting a very important correlation between layers. The use of all the bands simultaneously, just like $CO_2$, improves the information distribution along the vertical profile. The forward model error is larger than that of $CO_2$ since the methane is more affected by the interfering species. The smoothing error is significantly larger than $CO_2$, since it is constrained by a much higher a priori, which suggests a more direct effect on the retrieval of $CH_4$. Similarly to $CO_2$, when using a non-diagonal a priori covariance matrix, the vertical distribution is very analogous to that of the variance only. However, the a priori and a posteriori errors are significantly reduced. The total DOFS for $CH_4$ are shown in Table 5 for both SZA. This parameter shows that, for a diagonal a priori covariance matrix, three partial tropospheric columns can be retrieved, and one additional partial column for a SZA of 80°. Finally, while using a non-diagonal a priori covariance matrix the DOFS shows that one less partial column is retrieved.

As a general result, the simultaneous use of all the bands instead of using each one separately increases the total DOFS and reduces systematically the total errors of the two species. Moreover, using a climatological a priori covariance matrix shows the importance of reducing the error of the retrieved partial columns. Finally, the total profile error is derived from the relative values of the diagonal matrix of $S_x$ (see Tables 4 and 5), which are discussed in details in the following section.

### 3.4.2 Total column estimation and uncertainty

Ground-based instruments like the one used in the TCCON network and the EM27/SUN operate in the NIR, where the column-averaged dry-air mole fractions (denoted $X_G$ for gas G) is calculated by monitoring the observed $O_2$ columns. $X_G$ is calculated by rationing the gas retrieved slant column to the $O_2$ retrieved slant column for the same spectrum. Another method is used, especially among the NDACC community, to calculate $X_G$ without using the oxygen reference. Based on the formula given in Wunch et al. (2011) and used in Zhou et al. (2019), we can calculate $X_G$ for $CO_2$ and $CH_4$:

$$X_G = \frac{column_G}{column \, dry \, air} \tag{15}$$

$$column \, dry \, air = \frac{P_s}{g_{air} m_{air}^{dry}} - column_{H_2O} \frac{m_{H_2O}}{m_{air}^{dry}}$$

Where $m_{H_2O}$ and $m_{air}^{dry}$ are the mean molecular masses of water and dry air, respectively, $P_s$ the surface pressure and $g_{air}$ the column-averaged gravitational acceleration. Therefore, the calculation of $X_G$ is possible if all these parameters are available, particularly in the MAGIC framework where we have access, along with all the instruments involved, to the balloons and

radiosondes data (temperature, surface pressure, relative humidity, etc...). Thus, for these particular campaigns, $X_G$ values will be calculated for CHRIS using ARAHMIS, and the results will be compared with the other instruments involved, especially the IFS125HR of the TCCON network and the EM27/SUN, and this will be the subject of the upcoming paper. However, the two equations for the calculation of $X_G$ are not strictly similar since the EM27/SUN eliminates the systematic errors that are common to the target gas and $O_2$ column retrievals which will not be possible for us, since the $O_2$ band is not detected by CHRIS.

In addition, the total column uncertainty is calculated by summing the concentration of each layer along the profile weighted by the column of dry air based on figures 8 and 9. Table 2 lists the propagated uncertainties of the total column for both SZA using a diagonal a priori covariance matrix: the uncertainty of $CO_2$ total column is 2.89% and 2.6% for 10° and 80° respectively; while the uncertainty for $CH_4$ total column is 4.4% and 4.19% for 10° and 80° respectively. The uncertainties are smaller for a SZA of 80°, because the information distribution is improved with a longer OPD. Furthermore, these results show that the total profile error for $CH_4$ is almost 2 times higher than that of $CO_2$, but this is explained by the fact that our profile error is limited by the a priori profile error which is much higher for $CH_4$ than for $CO_2$. The dominating component of the uncertainty comes from the smoothing and predominates the other uncertainties for both GHG and is the major contributor to the total profile error. $H_2O$, temperature and SZA are the most important parameters contributing to the forward model, which are represented by the non-retrieved parameters uncertainty. Additionally, it is important to note that there is a supplementary uncertainty associated to the spectroscopy unaccounted for in our study, which is purely systematic. It's not simple to evaluate in this case, because we use different spectral domains each having different spectroscopic uncertainties listed in the HITRAN database.

| Error | $CO_2$ | | $CH_4$ | |
|---|---|---|---|---|
| SZA | 10° | 80° | 10° | 80° |
| Smoothing | 2.79 | 2.51 | 4.34 | 4.11 |
| Measurement | 0.6 | 0.54 | 0.59 | 0.7 |
| Non-retrieved parameters | 0.14 | 0.12 | 0.27 | 0.5 |
| Total | 2.89 | 2.6 | 4.4 | 4.19 |

Table 2: The total column errors for $CO_2$ and $CH_4$ profiles for CHRIS for the two SZA. The uncertainties are shown in percentages (%).

## 3.5 Comparison and complementary information content analysis for HR125, EM27/SUN and CHRIS

During the MAGIC campaigns, several EM27/SUN and two IFS125HR from the TCCON network were operated alongside CHRIS. An information content analysis is presented in the following sections for both of these instruments in order to compare and complement the study performed on CHRIS in Section 3.4.

### 3.5.1 Complementary information with EM27/SUN

In this section, an IC study is performed for the EM27/SUN instrument in order to compare it with our results and to investigate the possibility of complementing the data we obtained from CHRIS, especially for MAGIC. The bands of EM27/SUN used in this study are denoted as follows: B3, the common band with CHRIS, with a spectral range of 4700-5200 $cm^{-1}$, B4 going from 5460 $cm^{-1}$ to 7200 $cm^{-1}$, and B5 spanning the spectral region between 7370 and 12500 $cm^{-1}$.

Firstly, a similar study to CHRIS is performed on the EM27/SUN for $CO_2$ and $CH_4$ separately. As mentioned in Section 3.4, the state vector is constituted of only one of the gases concentrations with the same profile at a layer going from 0 to 40 Km; however, we took into account the SNR and spectral resolution specific to this instrument (see Table 3). Similar to the reasoning for CHRIS followed in Section 3.4, this study shows that using all the EM27/SUN bands together leads to an improvement of the a posteriori error profile on $CO_2$ concentrations, especially in the lower part of the atmosphere. Table 4 shows the DOFS for $CO_2$ of EM27/SUN: using a diagonal a priori covariance matrix for an angle of 10°, the total DOFS for bands B3 (common band with CHRIS), B4 and all bands together are: 2.95, 1.63 and 3.03 respectively. If only band B3 is taken into consideration which is the common band between the two instruments, the DOFS of CHRIS in this band is, as stated before, 2.62 and 3.34 for an angle of 10° and 80° respectively, compared to 2.95 and 3.17 for EM27/SUN. Therefore, the same number of partial columns can be retrieved using CHRIS (see Section 3.4) for $CO_2$ in this band. Furthermore, similarly to CHRIS, while using a non-diagonal a priori covariance matrix (Eguchi et al. (2010)), the total error is reduced with a more propagated smoothing error on the profile and a reduction in the DOFS. As for $CH_4$ and referring to Table 3, band 3 in this instrument begins (4700

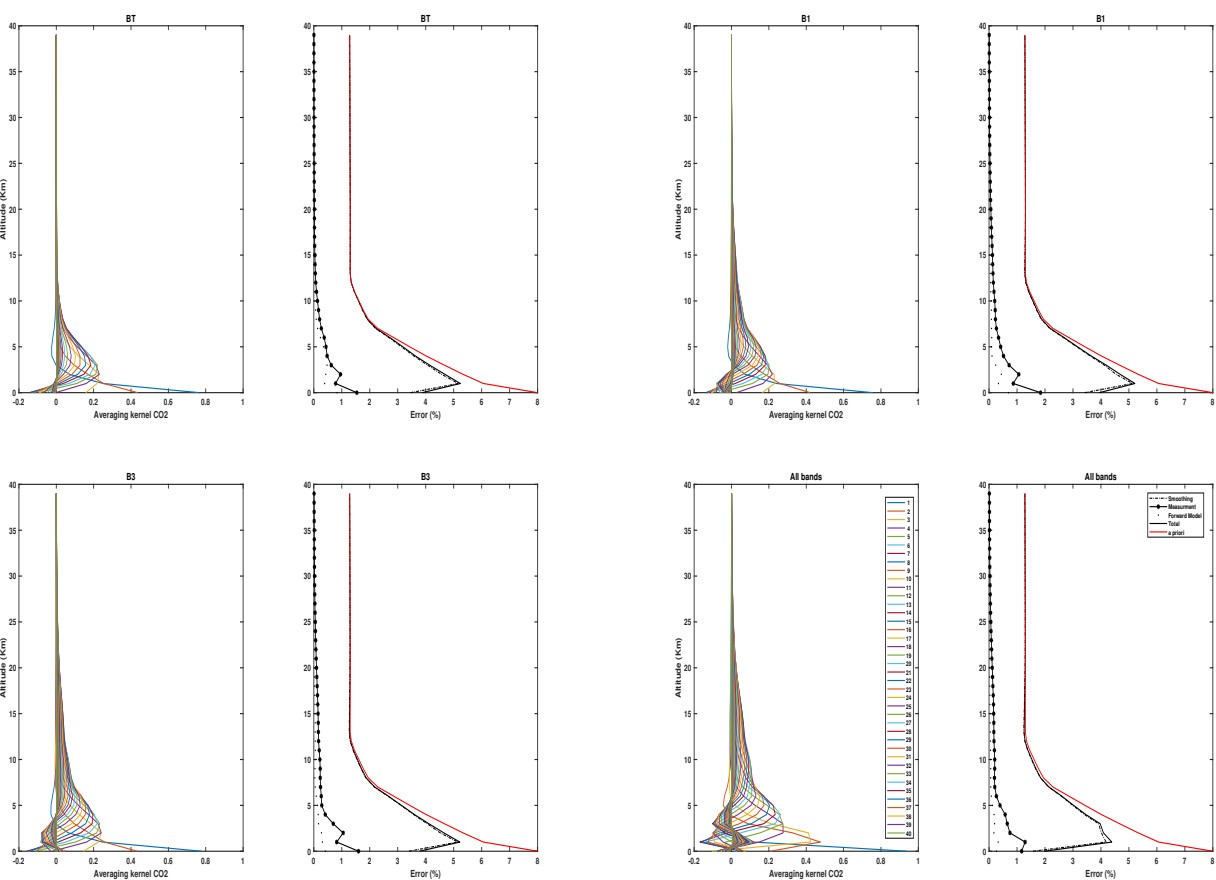

Fig. 8: Averaging kernels and error budgets of $CO_2$ vertical profiles for bands BT, B1 and B3 separately and all the bands together for an angle of $10°$ for CHRIS. The red and solid black lines stand for the prior ($S_a$) and posterior ($S_x$) errors respectively; the smoothing ($S_{smoothing}$), measurement ($S_{meas.}$), and forward model parameters ($S_{fwd.mod.}$) errors are dashed/dotted, dashed/starred and dotted, respectively.

| | Resolution ($cm^{-1}$) | MOPD ($cm$) | Spectral region ($cm^{-1}$) | SNR |
|---|---|---|---|---|
| CHRIS | 0.135 | 4.42 | BT: 680-1250 B1: 1800-2300 B2: 2400-3600 B3: 3900-5200 | 780 |
| EM27-SUN | 0.5 | 1.8 | B3: 4700-5200 B4: 5460-7200 B5: 7370-12500 | 1080 |
| IFS125HR (TCCON) | 0.02 | 45 | 4000-15000 | ~750 |
| IFS125HR (NDACC) | 0.0035-0.007 | 128-257 | 5-5200 | ~1000 |

Table 3: Instrumental characteristics of CHRIS, EM27-SUN and IFS125HR of both NDACC and TCCON.

$cm^{-1}$) where the $CH_4$ band ends (4150-4700 $cm^{-1}$) in IFS125HR and CHRIS. This is important because TCCON networks

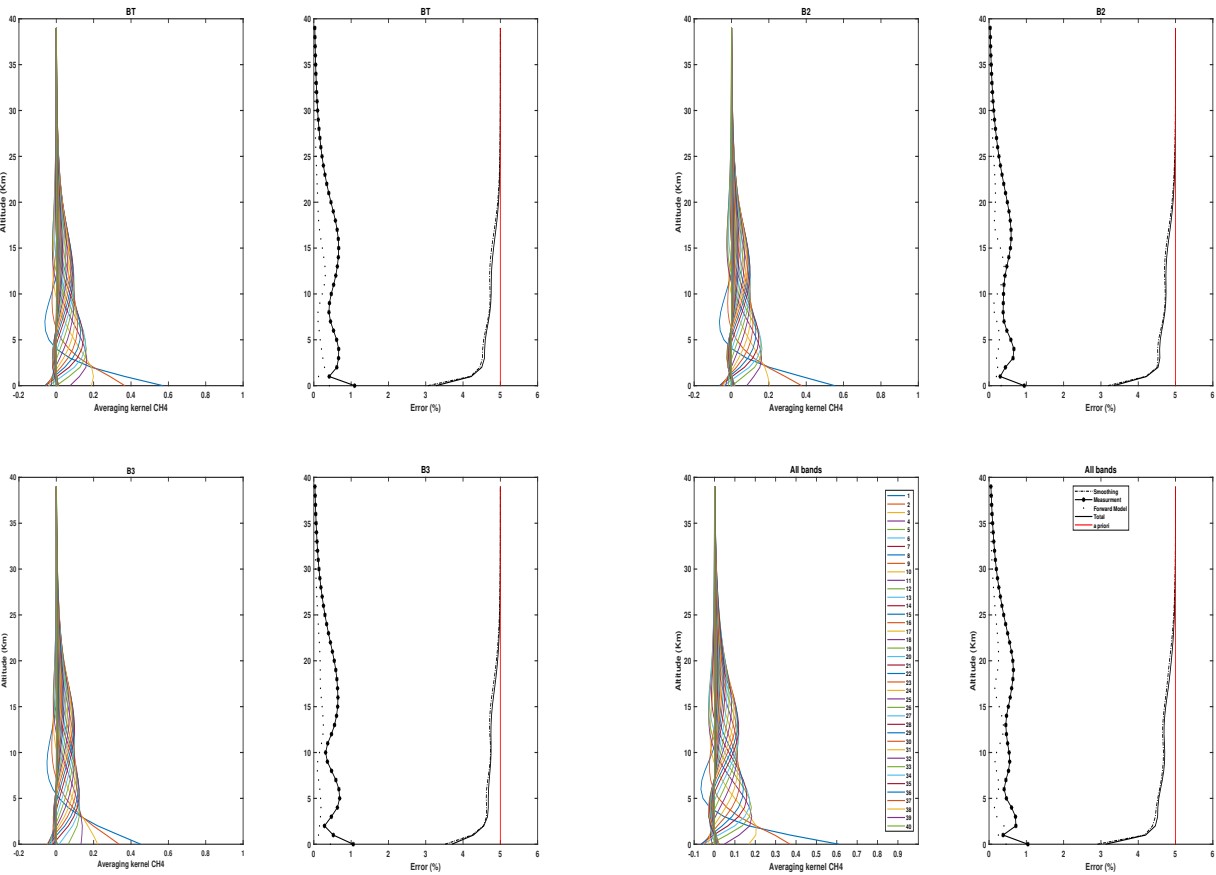

Fig. 9: Same as Fig. 8, but for $CH_4$ with bands BT, B2, B3 and all the bands together.

begin their measurements at 4000 $cm^{-1}$ that allowed us the comparison with the band 3 of CHRIS (for both $CO_2$ and $CH_4$). However, the EM27/SUN have no exploitable signal before 4700 $cm^{-1}$ (Gisi et al. (2012)), therefore the $CH_4$ absorption lines do not show in the common band between CHRIS and the EM27/SUN so the results are not discussed here.

440 Secondly, a simultaneous IC study was performed on all the channels of both CHRIS and EM27/SUN in order to analyze the complementary aspect of these two instruments. The results of this study are shown in Fig. 10. The DOFS obtained for $CO_2$ is 3.67 and 3.93 for angles 10° and 80° respectively; and for $CH_4$ 3.99 and 4.43. This indicates a significant improvement of the retrieval when the spectral synergy TIR/SWIR/NIR is used, but less than the one obtained from space (for example TANSO-FTS in Herbin et al. (2013a)), since the measurement is obtained from the same optical path.

445

### 3.5.2 Comparison with IFS125HR

As mentioned before, IFS125HR is a ground-based high resolution infrared spectrometer used in the NDACC and TCCON stations across the world. We performed a similar information content study only on the TCCON instrument since this network is involved in the MAGIC campaigns therefore the results can be compared. For simplicity, the same annotation of the bands is
450 kept for this section. The same methodology described in Section 3.4 is used here: the state vector is constituted of only $CO_2$ and $CH_4$ concentrations at a layer going from 0 to 40 km where the SNR and the spectral resolution specific to the IFS125HR are taken into consideration (see Table 3).

We follow the same reasoning as the sections before: Fig. 11 shows the averaging kernel $A$ and the total posterior error $S_x$ for $CO_2$ and $CH_4$, for an angle of 10°. We can see that the vertical distribution is more homogeneous than CHRIS and EM27/SUN
455 suggesting a high sensitivity at high altitudes although in the lower atmosphere, the a posteriori error $S_x$ is significantly reduced. This is also shown in the error budget study: we can still distinguish the a posteriori total error (solid black line) from the a priori

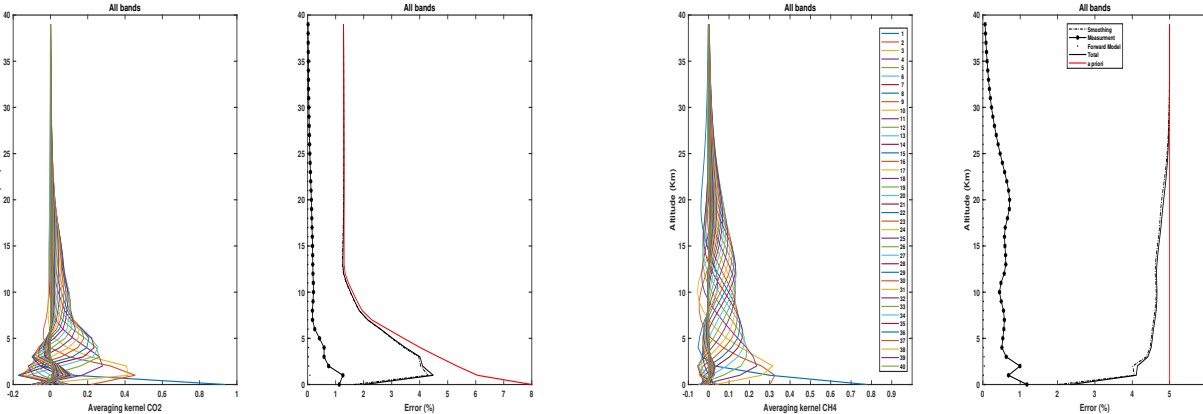

Fig. 10: Averaging kernels and error budgets of $CO_2$ and $CH_4$ vertical profiles for all the bands together for EM27/SUN and CHRIS combined for an angle of 10°. The red and black lines stand for the prior ($S_a$) and posterior ($S_x$) errors respectively; the smoothing ($S_{smoothing}$), measurement ($S_{meas.}$), and forward model parameters ($S_{fwd.mod.}$) errors are dashed/dotted, dashed/starred and dotted, respectively.

|  |  | BT | B1 | B3 | B4 | All bands | |
|---|---|---|---|---|---|---|---|
|  |  |  | | DOFS | | DOFS | Error |
| CHRIS | Angle 10 | 2.01 | 2.32 | 2.62 | - | 2.95 | 2.89% |
|  | Angle 80 | 2.56 | 2.67 | 3.34 | - | 3.71 | 2.6% |
| CHRIS with covariance | Angle 10 | 1.45 | 1.7 | 2.15 | - | 2.38 | 1.01% |
|  | Angle 80 | 1.89 | 1.92 | 2.68 | - | 3.08 | 0.94% |
| EM27/SUN | Angle 10 | - | - | 2.95 | 1.63 | 3.03 | 2.77% |
|  | Angle 80 | - | - | 3.17 | 2.33 | 3.31 | 2.67% |
| EM27/SUN with covariance | Angle 10 | - | - | 2.25 | 1.17 | 2.37 | 1.01% |
|  | Angle 80 | - | - | 2.53 | 1.71 | 2.68 | 0.97% |
| IFS125HR | Angle 10 | 2.15 | 2.33 | 3.07 | 2.62 | 3.9 | 2.82% |
|  | Angle 80 | 2.51 | 2.61 | 3.59 | 2.99 | 4.23 | 2.72% |
| IFS125HR with covariance | Angle 10 | 1.66 | 1.85 | 2.86 | 2.3 | 3.28 | 0.97% |
|  | Angle 80 | 1.97 | 2.14 | 3.04 | 2.61 | 3.53 | 0.95% |

Table 4: The DOFS and column errors (%) of $CO_2$ for each band and for each instrument.

| | | BT | B2 | B3 | B4 | All bands | |
|---|---|---|---|---|---|---|---|
| | | | DOFS | | | DOFS | Error |
| CHRIS | Angle 10 | 2.77 | 2.87 | 2.62 | - | 3.34 | 4.4% |
| | Angle 80 | 3.19 | 3.88 | 3.45 | - | 4.26 | 4.19% |
| CHRIS with covariance | Angle 10 | 2.03 | 2.22 | 1.97 | - | 2.57 | 1.5% |
| | Angle 80 | 2.23 | 2.83 | 2.56 | - | 3.21 | 1.46% |
| EM27/SUN | Angle 10 | - | - | - | 1.69 | 1.69 | 4.67% |
| | Angle 80 | - | - | - | 2.45 | 2.45 | 4.54% |
| EM27/SUN with covariance | Angle 10 | - | - | - | 1.18 | 1.18 | 1.59% |
| | Angle 80 | - | - | - | 1.81 | 1.81 | 1.55% |
| IFS125HR | Angle 10 | 3.37 | 3.97 | 3.69 | 2.56 | 4.64 | 4.23% |
| | Angle 80 | 3.66 | 4.42 | 4.23 | 3.35 | 4.98 | 4.21% |
| IFS125HR with covariance | Angle 10 | 2.45 | 3.03 | 2.79 | 1.83 | 3.55 | 1.47% |
| | Angle 80 | 2.57 | 3.34 | 3.22 | 2.45 | 3.81 | 1.46% |

Table 5: The DOFS and column errors (%) of $CH_4$ for each band and for each instrument.

error (red line) even in the higher atmosphere. This is explained by the fact that the IFS125HR has a spectral resolution higher than both CHRIS and EM27/SUN, so the measurement always improves our knowledge of the profile all along the atmospheric column. Also, when using a non-diagonal a priori covariance matrix, the total profile error is significantly reduced, especially for $CH_4$, however, the DOFS is also reduced.

The DOFS of $CO_2$ and $CH_4$, as shown in Table 4, for both viewing angles and a priori covariance matrices. On the one hand, one additional partial tropospheric column for $CO_2$ can be retrieved with respect to CHRIS for an angle of 10°, and EM27/SUN for both angles, if all the bands are used. On the other hand, one additional partial tropospheric column can be retrieved for $CH_4$ with respect to CHRIS, if all the bands are used and for both angles.

| | $CO_2$ | | $CH_4$ | |
|---|---|---|---|---|
| DOFS | 90% | 99% | 90% | 99% |
| Number of channels | 1329 | 4648 | 1387 | 5924 |
| % of the total number of channels | 2.15% | 7.54% | 2.25% | 9.61% |

Table 6: Corresponding number of selected channels for the DOFS of $CO_2$ and $CH_4$ and their respective percentage of the total number of channels for CHRIS.

# 4  Channel selection

Using all the channels in the retrieval process has two disadvantages. First of all, it requires a very large computational time. Secondly, the correlation of the interfering species increases the systematic error. In this case, the a priori state vector $x_a$ and the error covariance matrix $S_a$ is very difficult to evaluate. Channel selection is a method described by Rodgers (2000) to optimize a retrieval by objectively selecting the subset of channels that provides the greatest amount of information from high-resolution infrared sounders. L'Ecuyer et al. (2006) offers a description of this procedure based on the Shannon information content.

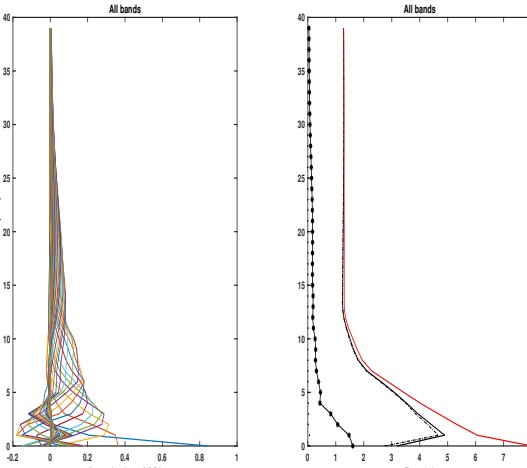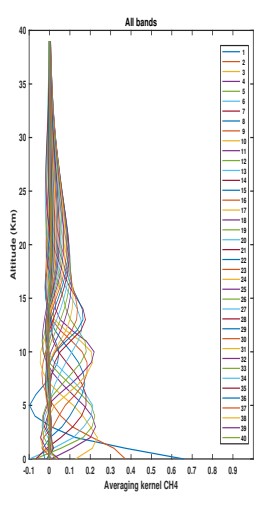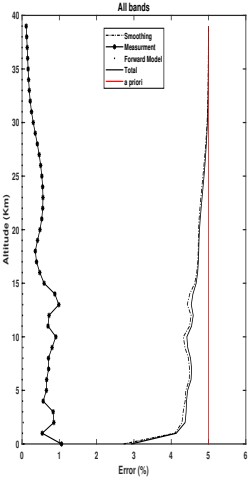

Fig. 11: Averaging kernels and error budgets of $CO_2$ and $CH_4$ and vertical profiles for bands all the bands together for an angle of 10° for IFS125HR. The red and black lines stand for the prior ($S_a$) and posterior ($S_x$) errors respectively; the smoothing ($S_{smoothing}$), measurement ($S_{meas.}$), and forward model parameters ($S_{fwd.mod.}$) errors are dashed/dotted, dashed/starred and dotted, respectively.

Firstly, we create an "information spectrum" in order to evaluate the information content with respect to the a priori state vector. The channel with the largest amount of information is then selected. A new spectrum is then calculated with a new posteriori covariance matrix that is adjusted according to the channel selected on the first iteration to account for the information it provides. In this way a second channel is chosen, based on this newly defined state space, that provides maximal information relative to the new a posteriori covariance matrix. This process is repeated and channels are selected sequentially until the information in all the remaining channels falls below the level of measurement noise. As stated by Shannon information content and noted in Rodgers (2000), it is convenient to work in a basis where the measurement errors and prior variances are uncorrelated in order to compare the measurement error with the natural variability of the measurements across the full prior state. Therefore, it is desirable to transform the Jacobian matrix $K$ (see Sect. 3.2) into $\widetilde{K}$ using:

$$\widetilde{K} = S_y^{-1/2} K S_a^{1/2} \tag{16}$$

which offers the added benefit of being the basis where both the a priori and the measurement covariance matrices are unit matrices. Furthermore, Rodgers demonstrates that the number of singular values of $\widetilde{K}$ greater than unity defines the number of independent measurements that exceed the measurement noise defining the effective rank of the problem.

Letting $S_i$ be the error covariance matrix for the state space after i channels have been selected, the information content of channel j of the remaining unselected channels is given by:

$$H_j = \frac{1}{2} log_2 (1 + \widetilde{k}_j^T S_i \widetilde{k}_j), \tag{17}$$

where $\widetilde{k}_j$ is the $j$th row of $\widetilde{K}$. $H_j$ constitutes the information spectrum from which the first channel is selected. Taking the chosen channel to be channel $l$, the covariance matrix is then updated before the next iteration using the following statement:

$$S_{i+1}^{-1} = S_i^{-1} + \widetilde{k}_l \widetilde{k}_l^T \tag{18}$$

In this way, channels are selected until 90% of the total information spectrum H is reached in a way that the measurement noise is not exceeded.

After that, H expressed in bits is converted to DOFS to obtain Fig. 12 that represents the $CO_2$ and $CH_4$ total DOFS evolution as a function of the number of selected channels and for all spectral bands and for a SZA of 10°. CHRIS has 75424 channels in total, 13800 are unusable because of the water vapor saturation between the bands, which leaves us with 61624 exploitable channels. A pre-selection of these channels, based on Fig.7, is done where the number of exploitable channels is reduced to the spectral areas where we find $CO_2$ and $CH_4$ (13447 and 19751 pre-selected channels respectively). At a first look at Figure

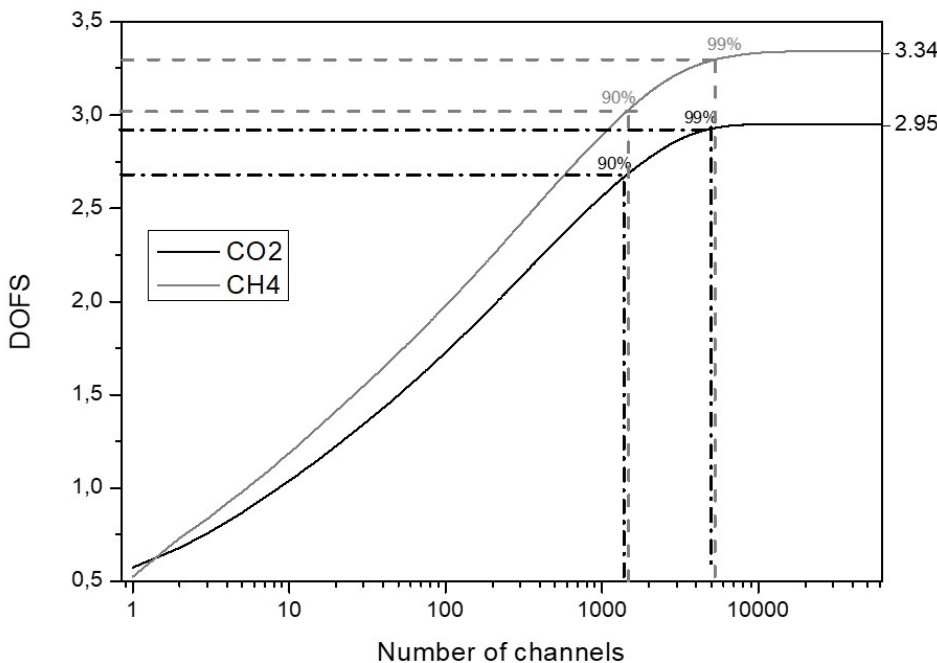

Fig. 12: Evolution of the DOFS with the number of selected channels for $CO_2$ (black) and $CH_4$ (gray).

12, the DOFS for each gas increases sharply with the first selected channels and then more steadily. The number of channels required to reach 90% and 99% of the total information is represented in Table 6. For $CO_2$, out of the 1329 channels, 55.76% of the information comes from B3 (common band with the EM27/SUN), 37.24% from BT and 6.99% from B1. As for $CH_4$, out of the 1387 channels, 46.86% of the information comes from B2, 28.19% from BT and 24.9% from B3. This result shows that most of the information for $CO_2$ and $CH_4$ comes from B3 and BT respectively, which indicates that the synergy between TIR and SWIR observations is confirmed.

Furthermore, the 1329 and 1387 selected channels represent 2.15% and 2.25% of the 61624 exploitable channels respectively, so a retrieval process that uses selected channels corresponding to 90% of the total information content would give comparable results to the one using the entire set of channels since almost 98% of the information is redundant. Hence, these results indicate the interest of determining an optimal set of channels for each gas separately, this is why this channel selection will be used in the retrieval process making it easier and less time consuming.

## 5 Conclusions

In conclusion, this paper presents the characteristics of the new infrared spectrometer CHRIS that allows the retrieval of GHG and trace gases. This instrumental prototype has unique characteristics, such as its high spectral resolution (0.135 $cm^{-1}$) and wide spectral range (680-5200 $cm^{-1}$) covering the MIR region. In the context of its exploitation to retrieve GHG, spectral and radiometric calibrations were performed using a calibrated external blackbody reaching a temperature of 1523 K. Additionally, between laboratory calibrations and during field campaigns the radiometric stability is monitored through measurements of the internal blackbody. In the MAGIC framework, an extensive information content analysis is performed showing the potential capabilities of this instrument to retrieve GHG, using two different SZA (10° and 80°) to quantify the improvement of the information with the solar optical path. Furthermore, two a priori covariance matrices were used: one diagonal and another derived from climatological data. The total column uncertainty is estimated showing that when using a diagonal a priori covariance matrix the error for an angle of 10° is of the order of 2.89% for $CO_2$ and 4.4% for $CH_4$ for all the bands; however, when using a climatological distribution the total column error for the same angle and for all the bands is reduced to 1.01%

for $CO_2$ and 1.5% for $CH_4$ but with a significant decrease in the DOFS (from 2.95 to 2.38 for $CO_2$ and from 3.34 to 2.57 for $CH_4$). Also, a comparison study with the IFS125HR of the TCCON, which is widely used in the satellite validation process, is performed illustrating the benefits of its high spectral resolution on GHG retrievals. Moreover, a complementary study is carried out on the EM27/SUN to investigate the possibility of a retrieval exploiting the synergy between TIR/SWIR/NIR observations, which showed that a significant improvement can be obtained, for example with a SZA of 10° the DOFS is increased from 2.95 to 3.67. Finally, a channel selection is implemented to remove the redundant information. The latter will be used in the future work dedicated to the $CO_2$ and $CH_4$ total columns retrievals for the MAGIC campaigns.

*Acknowledgements.* The CaPPA project (Chemical and Physical Properties of the Atmosphere) is funded by the French National Research Agency (ANR) through the PIA (Programme d'Investissement d'Avenir) under contract "ANR-11-LABX-0005-01" and by the Regional Council "Nord Pas de Calais-Picardie" and the "European Funds for Regional Economic Development" (FEDER). Financial support from CNES TOSCA (MAGIC) is also acknowledged.

Also we acknowledge the "Ecole centrale de Lille" for its help in providing the cavity blackbody used in the calibration process. We express our appreciation for the Pr. Denis Petiprez for his help with the determination of the ILS. Finally, we are also immensely grateful to Fabrice Ducos who provided technical expertise that greatly assisted the research and for his work on ARAHMIS.

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
