# Peer review of "Instrumental characteristics and potential Greenhouse gases measurement capabilities of the Compact High-spectral Resolution Infrared Spectrometer: CHRIS."

_Atmospheric Measurement Techniques, 2019_

## Referee Comment (RC1) · Anonymous Referee #1 · 21 Nov 2019

Please see the attached pdf file.

Please also note the supplement to this comment:
https://www.atmos-meas-tech-discuss.net/amt-2019-391/amt-2019-391-RC1-supplement.pdf

---

## Referee Comment (RC2) · Anonymous Referee #2 · 13 Jan 2020

The paper introduces a new portable Fourier Transform Spectrometer (FTS) with a higher resolution than other commercially available portable FTS instruments. The characteristics of the instruments are described in detail and methods for spectral and radiometric calibration are implemented. The information content to retrieve the vertical profile information of $CO_2$ and $CH_4$ and the associated errors are explained and compared to two other FTS instruments that are widely used within the remote sensing community. The optimized number of channels to retrieve $CO_2$ and $CH_4$ are carried out at the end.

[Figure]

The work presented in the manuscript is within the scope of AMT. The text sometimes becomes unclear and hard to follow. I suggest reviewing the transitions between topics and fleshing out when a new topic is introduced to make it easier for the readers to follow along.

Here are some general comments:

1. The MAGIC campaign has been brought up a few times in the manuscript, however, it's unclear when exactly that campaign took place? How many days of measurement are available from each instrument involved?

2. The title of the manuscript suggests "greenhouse gas measurement capabilities" of the instrument are discussed in the paper. However, the main focus of the paper is on the information that could be obtained for the vertical profiles of $CO_2$ and $CH_4$ and there's no presentation of retrieved column values. Given that two other instruments (125-HR and EM27/SUN) were measuring at the same time as CHRIS, comparison of $CO_2$ and $CH_4$ column values between the three instruments could be proving the "greenhouse gas measurement capabilities" of CHRIS. GGG could easily used perform the retrievals.

3. In section, 2.2.1 it is mentioned that 50, 100 spectra are coadded for $CO_2$ and $CH_4$ measurements. My calculations using the laser frequency and scanner velocity suggests a single scan time of about 0.7 s. Can you confirm this number? If that's the case, caoadding 100 spectra is still fast enough not to worry about changes in the atmosphere and also stability of the laser.

4. Although $H_2O$ absorption lines are present in almost all spectral windows, water vapour mole fractions are not retrieved in the analysis. Is it because of the certain meteorological conditions in Izaña that leads to stable water vapour values? Bringing some evidence to prove that's the case would be helpful.

5. In section 3.1, it is mentioned the apriori profiles (I am guessing of $CO_2$ and $CH_4$), temperature and humidity are used for the analysis. Could you please specify where these information are obtained from?

6. Page 13, the last paragraph, you mention that calculation of $X_G$ using $O_2$ column values done by EM27/SUNs allows comparisons with satellite data and it's not possible for CHRIS. This statement contradicts the point made earlier in the introduction where you suggest CHRIS could be used for satellite validation. In fact, retrieval of $X_G$ values are possible for CHRIS if surface pressure and water vapour measurements are used as described by Wunch et al., 2010.

Minor corrections and comments:

- Please use EM27/SUN all over the text as EM27 might be mistaken by the other Bruker instrument.

- Line 18: I think you mean InfraRed High Spectral Resolution "Spectroscopy"

- Line 29: There are currently more than 30 125HRs running if you check https://tccon-wiki.caltech.edu/Sites and http://www.ndaccdemo.org/stations.

- Section 2.2.3: could you specify by a subtitle to show which parts are about spectral calibration and which parts are for radiometric calibration?

- Line 135: "This reference wavelength...." Consider rewording this sentence. Changes in pressure and temperature cause the refractive index to change and as a consequence the reference wavelength will change.

- Line 157: Could you specify which two temperatures you used for the black body for calibration?

- Figure 4: I suggest you separate each panel by a frame and label it with the name of the corresponding band.

- Figures 5,6,8 and 9: Please add a legend to specify the colors for the averaging kernel plots.

- Line 322: The same number of columns? Do you mean the same number of vertical layers?

- Line 325: This is not quite right. $CH_4$ for both 125HR and EM27/SUN are retrieved from microwindows around 6000 cm-1.

- Line 383: Define K.

- Section 4: You specify the number of channels used for each species can you also specify where the centre of the band is?

- Figure 10: caption. $CH_4$ (green line).
* * *

---

## Author Comment (AC1) · 21 Feb 2020

**Reply anonymous reviewer #1**

The publication under consideration introduces a new portable spectrometer commercially available from Bruker, which offers moderate spectral resolution (max optical path difference 4.42 cm) and wide spectral coverage (680 – 5200 cm$^{-1}$). The authors present the instrumental setup, the methods applied for achieving radiometric calibration, and finally present a theoretical study on the presumed instrumental capabilities.
The topics addressed by the paper match well with the scope of AMT. However, in my impression, the actual elaboration falls short with respect to the premises stated in the title, so major extensions (and a readjustment of the title?) are required before the work can be considered for publication in AMTD.
The title phrase "Instrumental characteristics" would suggest that real world instrumental characteristics are elaborated and reported, what currently is reported hardly reaches beyond the specifications provided by the manufacturer.

We thank the anonymous reviewer for his careful reading of our manuscript and his many insightful comments and suggestions. After careful consideration, we are providing a revised manuscript that benefits from further calculation results, figures and rewriting of certain sections that will be discussed in details in the following answers. This revised manuscript reflects the recommendations of the reviewer, and it benefits from a slightly modified title, as suggested by the reviewer: **Instrumental characteristics and potential Greenhouse gases measurement capabilities of the Compact High-spectral Resolution Infrared Spectrometer: CHRIS**. Due to these modifications, we have considerably improved the manuscript.

First, we want to point out again that CHRIS is an instrumental prototype, and that the first part of the manuscript is dedicated to the characterization of its main characteristics that have not been provided by the manufacturer. Secondly, the potential capabilities to measure greenhouse gases are presented in the second part of the manuscript through an exhaustive information content study and a comparison with two other commercial FTIR: the EM27/SUN and the IFS125HR.

Below we respond to the questions and comments of the reviewer in detail, with reviewer comments in a different color.

1- Which SNR is achieved in the measured spectrum as function of wave number (this also requires the specification of scan speed and total scan time applied, which is not reported in the current version of the draft)?

During the acquisition of a spectrum, the amount of the signal received by the detector depends on the aperture stop diameter, which can be manually modified when the instrument is open. Since this manipulation is not easy during measurements campaigns outside the laboratory, we performed several tests in different conditions and we found that an aperture stop of 5 mm is the best compromise between a good signal and the saturation of the detector. We adopted this diameter for all the spectra recorded by CHRIS.

Furthermore, several tests allowed us the quantifying of the Signal-to-Noise Ratio (SNR) improvement with the scan number (if we increase the number of scans, we get a higher SNR, as can be shown in **Fig. 1**). However, we can see that there is no

apparent improvement between the spectra of 100 and 200 scans, and this is why we limited our number of scans to 100 for measuring $CO_2$ and $CH_4$. Moreover, the time needed for 1 scan with a scanning velocity of 120 KHz is 0.83s, so 100 scans take approximately 83s which is low in comparison with the variability of $CO_2$ and $CH_4$ in the atmosphere.

[Figure]

**Figure 1:** Improvement of the SNR with different scan number and a scan speed of 120 KHz

A typical example of the SNR of such a protocol (aperture stop diameter= 5 mm, scan speed=120 KHz and scan number=100) is shown in **Fig. 2**. The SNR varies between 20 and 845 in the covered spectral domain.

[Figure]

**Figure 2:** The SNR in function of the wavenumber for CHRIS.

These information and the figures have been added to the revised manuscript.

2- How stable are the radiometric characteristics of the device (use e.g. Allan deviation plots of spectral or derived quantities, see e.g. the investigation performed on a portable spectrometer by Chen et al., 2016)? Characterize the instrumental line shape (ILS) by using lasers, gas cells or open path measurements (see, e.g. Frey et al., 2015). Is the ILS near the nominal expectation? Does the spectrometer achieve the nominal resolution at all?

The cavity blackbody used is an HGH/RCN1250N2 certified by the LNE (Laboratoire National de métrologie et d'Essais) as having an emissivity greater than 0.99 in the spectral covered domain of CHRIS, a stability of 0.1° K at 1173° K, an opening diameter up to 50 mm (corresponding to that of CHRIS), and covering temperatures from 323 to 1523° K. This cavity blackbody is mounted on an optical bench and used before and after each campaign to perform absolute radiometric calibrations through open path measurements and make sure that this calibration is stable across the spectral range covered.

Furthermore, during campaigns and after long transportations, constant measurements of the internal blackbody, which can be heated up to 353° K, is carried out in the thermal infrared region that is the most affected by any drift. **Figure 3** shows the variations of the internal blackbody during multiple field campaigns: we can see a little fluctuation in function of the measurement conditions, but we can clearly see that depending on the locations and even years, no systematic drift can be detected, so we can safely say that the instrument is quite stable between each laboratory calibration. We give more details in the revised manuscript.

[Figure]

**Figure 3:** The radiometric stability of the instrument is achieved by following the variations of the internal blackbody during multiple field campaigns. IZA : Izana, MBR : M'Bour, VDA : Villeneuve d'Ascq, ASA : Aire-sur-l'Adour.

Since CHRIS is a prototype, there is no nominal expectation other than having a resolution at least similar or better than the future satellite sounders, e.g. IASI-NG (0.25 cm$^{-1}$). The tests conducted by the manufacturer confirmed that the spectral resolution is better than 0.2 cm$^{-1}$, which matches the original request.

Regarding the ILS, no information was provided upon delivery of the instrument, and since we were not equipped with the proper tools, there was no detailed information dedicated to the characterization of the ILS. Nevertheless, with the help of colleagues from the PC2A laboratory, we performed one open path ILS measurement alongside the pre-spectral calibration by using our cavity blackbody as source as it is previously done in Wiacek et al. 2007. We recorded the spectrum without applying any apodization, and with the help of the radiative transfer algorithm, ARAHMIS, developed at our laboratory we determined the ILS corresponding to a MOPD of 4.42 cm and a spectral resolution of 0.135 cm$^{-1}$ using a sinc function, which matches our expectations.

However, in order to characterize the ILS more regularly, we are in the process of modifying our laboratory optical bench to include gas cell measurements before the next field campaign which will take place this summer. Note that the ILS parameters can be easily modified in our RT code ARAHMIS, during the retrieval process.

3- On which levels other well-known important kinds of spectral artefacts are controlled which might critically detoriate retrieval quality? For a broadband spectrometer covering more than one octave along the frequency axis non-linearity, double passing and sampling ghosts need to be investigated.

The following information have been added to the revised manuscript:

There are commonly several well-known spectral artifacts: the aliasing, the picket-fence effect also known as the resolution bias error, and the phase correction, which are controlled in CHRIS. Several tests were performed on this instrument to investigate its optical and technical properties and the results are presented below.

An aliasing is the result of the missampling of the interferogram in the x-axis locations, which leads to errors in the retrieved column abundances due to its overlap with the original spectrum. Sampling positions are derived from the zero crossings of a He-Ne laser wave having a wavenumber of 15800 cm$^{-1}$. As a zero crossing occurs every $\lambda/2$, the minimum possible sample spacing is $\Delta x = 1/31600$ cm. This corresponds to a folding wavenumber ($\nu_f = 1/(2.\Delta x)$) of 15800 cm$^{-1}$, i.e. the maximum bandwidth that can be measured without overlap has a width of 15800 cm$^{-1}$. This source error is of special relevance to the spectra acquired in the NIR region. However, for the MIR, the investigated bandwidth is much smaller than 15800 cm$^{-1}$, where $\nu_{max}$ is less than 5200 cm$^{-1}$ for CHRIS's spectra so they are not affected by this problem (Dohe et al. 2013).

The picket-fence effect, or the resolution bias error, becomes evident when the interferogram contains frequencies, which do not coincide with the frequency sample points, but this is overcome in our spectra by the classical method of the Zero Filling Factor (ZFF) where the number of points is doubled in the original interferogram.

The phase correction is necessary while converting the interferogram into a spectrum, which is relevant to single-sided measurements, similar to those acquired by CHRIS since its interferometer cannot operate in a double-sided mode. Mertz phase correction is the method used in CHRIS to overcome this problem.

During our tests, we noticed that when using a scan speed of 160 KHz, we drastically increase the non-linearity effect of the detector (**Fig. 4**). However, we identified a ghost signal for low scanning velocities (for example 40 KHz shown in **Fig. 5**). This ghost is specific to CHRIS because it is caused by the noise introduced from the vibrations of the compressor used in the closed-cycle stirling cooler. The choice of a scanning velocity of 120 KHz is a compromise between two important features: the elimination of the ghost signal, which appears at scanner velocities below 80 KHz, and the increase of the detector non-linearity at a velocity of 160 KHz.

[Figure]

**Figure 4:** Spectra of the external cavity blackbody with two different scanning velocities.

[Figure]

**Figure 5:** Spectra of the internal blackbody of CHRIS with a scanner velocity of 40 KHz.

4- The spectrometer offers higher spectral resolution than other portable units described earlier by other investigators. Nevertheless, it still uses a similar non-stabilized He-Ne laser (which might drift during recording interferograms). Which recommendations concerning applied scan speed result from this? How many interferograms can be coadded before spectral recalibration is needed in order to avoid degradation of spectral resolution?

As said before (see answer 1), several tests were performed to find the best compromise of the above-mentioned parameters to achieve good spectral quality. For the covered spectral domain, one recorded interferogram consists of 99527 points with a scanner velocity of 120 KHz, which corresponds to a scan time of 0.83 s. The protocol was chosen in a way that when co-adding 100 interferograms, the total time required to measure a spectrum is 83 s, which is low in comparison with the variability of these gases in the atmosphere, so spectral degradation is avoided. Furthermore, the spectral parameters are adjustable at each inversion and for each spectrum in the RT model ARAHMIS.

The title phrase "Greenhouse gases measurement capabilities" would suggest that some kind of empirical validation and verification with respect to other independent observation systems is performed. A high level of precision and accuracy needs to be achieved (in the sub percent range), otherwise remotely-sensed measurements of column-averaged abundances of carbon dioxide or methane are essentially useless due to the low variability of these long-lived gases in the atmosphere. The TCCON was the first network to solve this task for ground based solar absorption spectroscopic observations. The successful strategy was to extend spectral observations into the near infrared including the 1.26 um band of molecular oxygen, so transferring the absolute measurement of a column amount into a relative measurement of the ratio of the target gas column to the column of molecular oxygen. Target gas and oxygen column amounts are derived from the same spectrum and the fact that the dry molar fraction of molecular oxygen is well known can be exploited. This strategy reduces the impact of many instrumental and other detrimental effects. It is difficult to see how a spectrometer not using the oxygen reference can match the abovementioned requirements. For demonstrating that the quality of data collected with CHRIS nevertheless is sufficient for validation and other purposes, a long-term side-by-side empirical test next to a TCCON spectrometer would be required. The period of investigation should cover at least one year for demonstrating the measurements can reproduce the annual cycle of the column-averaged abundances and for investigating possible biases related to atmospheric humidity, solar elevation, and other possible impact factors. Stability characteristics of the portable EM27/SUN spectrometer have been investigated by Frey et al., 2018 (measuring three years side-by-side to a TCCON spectrometer). First results of a long-term study performed at the TCCON site Sodankyla encompassing different kinds of portable spectrometers have recently been presented by Sha et al. (currently handled in AMTD: AMT-2019-371).

Indeed, we agree that measuring the oxygen column allows the determination of the carbon dioxide column where systematic effects are eliminated (Wunch et al.

2011). However, there is another option for calculating the column-averaged dry mole fraction without using the oxygen reference. Based on the formula given in Wunch et al. (2010) and used in Zhou et al. (2018), we can calculate $X_G$ for $CO_2$ and $CH_4$:

$$X_G = \frac{column_G}{column\ dry\ air}$$

$$column\ dry\ air = \frac{P_s}{\{g\}_{air} m_{air}^{dry}} - column_{H_2O} \frac{m_{H_2O}}{m_{air}^{dry}}$$

Where $m_{H_2O}$ and $m_{air}^{dry}$ are the mean molecular masses of water and dry air, respectively, $P_s$ is the surface pressure and $\{g\}_{air}$ is the column-averaged gravitational acceleration. Therefore, the calculation of $X_G$ is possible if all these parameters are available, particularly in the MAGIC framework where we have access, along with all the instruments involved, to the balloons and radiosondes data (temperature, surface, pressure, $H_2O$ vmr, etc…). However, the two equations for the calculation of $X_G$ are not strictly similar since the EM27/SUN eliminates the systematic errors that are common to the target gas and $O_2$ column retrievals. Addressing this issue, this paragraph is rewritten in the revised manuscript to eliminate any ambiguity.

We would like to mention that during MAGIC, CHRIS also performed simultaneous measurements co-located spatially and temporally with the EM27/SUN and the IFS125HR of the TCCON network, and the retrieval process and the calculation of $X_G$ using ARAHMIS is the subject of the study that had just begun and will be presented in an upcoming paper.

Judging from these studies and the experiences gained by TCCON, main obstacles in achieving the desired quality level are all kinds of practical instrumental issues and various details of the data processing, which all need to be addressed. The argumentative connection between the schematic theoretical information content analysis presented by the authors and the actual "greenhouse gases measurement capabilities" of CHRIS therefore seems weak. The theoretical study can only provide a rather optimistic theoretical extrapolation of the instrument's capabilities, and would suggest a modified title as "…and potential Greenhouse gases measurement capabilities …" or alike.

We agree that we do not present direct measurement capabilities of $CO_2$ and $CH_4$, so the main objective of this study is to investigate the potential capabilities of this instrumental prototype to measure greenhouse gases, hence the adjustment of the title as suggested by the reviewer. Nevertheless, in order to have results that are more reliable, we added to the revised manuscript the same information content study but using covariance matrices taken from TCCON data and close to those determined in Eguchi et al. 2010.

I would recommend adding substantial empirical evidence concerning the capabilities of measuring greenhouse gases, especially as the spectrometer was already operated side-by-side to a TCCON spectrometer on Tenerife island and I assume together with other instruments and aircore soundings during MAGIC. I understand that the data analysis scheme for CHRIS is still under development, but well-characterized codes for the processing as GFIT are available and could be used meanwhile.

Indeed, as mentioned before this substantial empirical evidence of the real capacities of the instrument is the main objective of the work that had just begun, within the framework of MAGIC, on the retrieval process of CHRIS. The latter is processed alongside different types of instrument: the ground-based commercial instruments like the IFS125HR from TCCON and the EM27/SUN; but also the methane lidar, the Aircore and the Amulse, and will therefore be the subject of the upcoming paper. We will use for these retrievals the radiative transfer model ARAHMIS, which has the particularity to apply the retrieval of absolute radiances on different spectral regions simultaneously, including the thermal band.

5- Under Equation 5 it is stated that the forward model F takes into account surface emissivity – not a good example in case of an upward looking solar absorption spectrometer as considered here:

Surface emissivity is not considered in the forward model F. It is a mistake which is corrected in the revised manuscript.

6- Line 235: the model atmosphere extends only up to 20 km altitude? This neglects a non-negligible fraction of the total column for e.g. carbon dioxide:

The DOFS of $CO_2$ at a zenithal angle of 10° at 40 km is 3.54 compared to 3.49 at 20 km, which shows that we have a weak sensitivity above 20 km. Furthermore, the profiles acquired by the instruments on-board the planes and balloons during MAGIC extend to 25 km, so we took into consideration the reviewer advice and we extended our study to 40 km to match the altitudes reached by TCCON as seen in **Figure 6**. All the new DOFS and figures are corrected accordingly in the revised manuscript.

[Figure]

**Figure 6:** The averaging kernel for $CO_2$ of CHRIS from 0 to 40 km in altitude.

7- Line 240: The assumption of zero off-diagonal elements in the a-priori covariance for long-lived greenhouse gases is highly unrealistic. Climatological data describing the variability of greenhouse gases profiles are available and should be used.

One of the main objectives of the acquisition of CHRIS is the validation of the space instruments, like TANSO-fts/ GOSAT, which has similar spectral bands. In a previous study (Herbin et al. 2013), diagonal a-priori covariance matrices were used, so for the sake of comparison we used the same matrices. Moreover, the use of diagonal a-priori covariance matrices is common for the retrievals from space measurements (e.g. De Wachter et al. 2017).

However, in our retrieval process we will use the more realistic a priori covariance matrix mentioned before. In the revised paper, the study with the two matrices is shown.

8- Line 255: Handling water vapor as a non-retrieved parameter does not match with retrieval strategies as applied in practice. Water vapor will be part of the retrieved state vector if in the spectral range under consideration it is not negligible as interfering species:

As mentioned before, since this information content study has to be easily compared to measurements from space instruments, we considered variability for the water vapor profile as derived from the IASI level 2 products provided by EUMETSAT (Herbin et al. 2013, De Wachter et al. 2017). However, we agree that the impact of water is very important, and in the work that had just begun on the $CO_2$ retrieval, $H_2O$ is part of the retrieved state vector. This information is mentioned in the revised manuscript.

9- Table 2 is inconsistent. Either "IFS125HR" refers to a TCCON spectrometer, then 45 cm MOPD is correct or it refers to an IFS125HR spectrometer operated by NDACC, in this case the 0.005 $cm^{-1}$ resolution is reasonable. The specified spectral range is incorrect for either network. From the viewpoint of an information content analysis it might be preferable to separate the two kinds of spectrometers, the FTIR spectrometers operated in the framework of NDACC and those operated by TCCON.

The information content analysis in the revised manuscript takes into consideration the characteristics of the two networks and therefore we separated our study to respect the characteristics of each network.

10- Line 325: Both TCCON and EM27/SUN (dual-channel version) spectrometers offer a lower wavenumber limit of about 4000 $cm^{-1}$:

We agree that this sentence is unclear, therefore in the revised manuscript it is replaced by the following: "… the TCCON network begin their measurements at 4000 $cm^{-1}$ that allowed us the comparison with the band 3 of CHRIS (for both $CO_2$ and $CH_4$). However, the EM27/SUN have no exploitable signal before 4700 $cm^{-1}$ (Gisi et al. 2012), therefore the $CH_4$ absorption lines do not show in the common band between CHRIS and the EM27/SUN…"

**The following references have been added to the revised manuscript:**

Wiacek, A., Taylor, J. R., Strong, K., Saari, R., Kerzenmacher, T. E., Jones, N. B., Griffith, D. W. T.: Ground-Based Solar Absorption FTIR Spectroscopy: Characterization of Retrievals and First Results from a Novel Optical Design Instrument at a New NDACC Complementary Station, Journal of Atmospheric and Oceanic Technology, 10.1175/JTECH1962.1.

Dohe, S., Sherlock, V., Hase, F., Gisi, M., Robinson, J., Sepúlveda, E., Schneider, M., and Blumenstock, T.: A method to correct sampling ghosts in historic near-infrared Fourier transform spectrometer (FTS) measurements, Atmos. Meas. Tech., 6, 1981–1992, https://doi.org/10.5194/amt-6-1981-2013, 2013.

Eguchi, N., Saito, R., Saeki, T., Nakatsuka, Y., Belikov, D., and Maksyutov, S.: A priori covariance estimation for $CO_2$ and $CH_4$ retrievals, J. Geophys. Res., 115, D10215, doi: 10.1029/2009JD013269, 2010.

De Wachter, E., Kumps, N., Vandaele, A. C., Langerock, B., and De Mazière, M.: Retrieval and validation of MetOp/IASI methane, Atmos. Meas. Tech., 10, 4623–4638, https://doi.org/10.5194/amt-10-4623-2017, 2017.

---

## Author Comment (AC2) · 21 Feb 2020

**Reply anonymous reviewer #2**

The paper introduces a new portable Fourier Transform Spectrometer (FTS) with a higher resolution than other commercially available portable FTS instruments. The characteristics of the instruments are described in detail and methods for spectral and radiometric calibration are implemented. The information content to retrieve the vertical profile information of $CO_2$ and $CH_4$ and the associated errors are explained and compared to two other FTS instruments that are widely used within the remote sensing community. The optimized number of channels to retrieve $CO_2$ and $CH_4$ are carried out at the end.
The work presented in the manuscript is within the scope of AMT. The text sometimes becomes unclear and hard to follow. I suggest reviewing the transitions between topics and fleshing out when a new topic is introduced to make it easier for the readers to follow along.

We thank the anonymous reviewer for his careful reading of our manuscript. After thorough consideration, we are providing an improved manuscript that reflects his insightful suggestions and comments with a slightly modified title as follows: **Instrumental characteristics and potential Greenhouse gases measurement capabilities of the Compact High-spectral Resolution Infrared Spectrometer: CHRIS.** As recommended, this revised manuscript benefits from clearer transitions between topics and rewriting some of sentences/sections, as well as further calculation results and figures that will be discussed in details in the following answers.

Below we respond to the questions and comments of the reviewer in detail, with reviewer comments in a different color.

1- The MAGIC campaign has been brought up a few times in the manuscript, however, it's unclear when exactly that campaign took place? How many days of measurement are available from each instrument involved?

The MAGIC campaigns are annual campaigns held at different locations in France (see **Fig. 1**), with the collaboration of different French laboratories: LMD, GSMA, LERMA, LSCE, and the LOA. In the future, these campaigns will be organized once or twice a year: once in France and once at the stratospheric balloon release sites (Kiruna, Sweden and Timmins, Canada). Up until now, there are about ten days of data available for each instrument involved during the last two campaigns. The objective is the monitoring of the emission of GHG, mainly $CO_2$ and $CH_4$, and to provide regular mobile data for the validation of the current and future space missions, like Merlin, Microcarb and IASI-NG.

We want to point out again that CHRIS is an instrumental prototype, and that MAGIC offers the ideal framework to its characterization, which motivated the first part of the manuscript. The potential capabilities to measure greenhouse gases are presented in the second part of the manuscript through an exhaustive information content study and a comparison with two other commercial FTIR: the EM27/SUN and the IFS125HR. However, the context of these campaigns and the scientific analysis of the measurements made by CHRIS and the inter-comparisons with the different instruments involved, will be detailed in an upcoming paper. Some extra details on MAGIC are mentioned in the revised manuscript.

[Figure]

**Fig 1:** CO$_2$ vertical concentration and instrument deployment for the MAGIC campaigns: a) 2018 and b) 2019

2- The title of the manuscript suggests "greenhouse gas measurement capabilities" of the instrument are discussed in the paper. However, the main focus of the paper is on the information that could be obtained for the vertical profiles of CO$_2$ and CH$_4$ and there's no presentation of retrieved column values. Given that two other instruments (125-HR and EM27/SUN) were measuring at the same time as CHRIS, comparison of CO$_2$ and CH$_4$ column values between the three instruments could be proving the "greenhouse gas measurement capabilities" of CHRIS. GGG could easily be used perform the retrievals.

     Indeed, we agree that the objective of this paper is the investigation of the potential capabilities of this instrument to measure the greenhouse gases, hence the adjustment of the title. The real capacities of this instrument and the preliminary results of the retrieval process is the main objective of the work that had just begun on CHRIS, within the framework of MAGIC. The latter is processed alongside different types of instruments: the ground-based commercial instruments like the IFS125HR from TCCON and the EM27/SUN; but also a methane lidar, the Aircore and the Amulse, and will therefore be the subject of the upcoming paper. However, we will use for these retrievals the radiative transfer model ARAHMIS, which has the particularity to apply the retrieval of absolute radiances on different spectral regions simultaneously, including the thermal band.

3- In section, 2.2.1 it is mentioned that 50, 100 spectra are coadded for CO$_2$ and CH$_4$ measurements. My calculations using the laser frequency and scanner velocity suggests a single scan time of about 0.7 s. Can you confirm this number? If that's the case, caoadding 100 spectra is still fast enough not to worry about changes in the atmosphere and also stability of the laser.

     Indeed, for the covered spectral domain, one recorded interferogram consists of 99527 points with a scanner velocity of 120 KHz, which corresponds to a scan time of 0.83 s. After several tests on the scan velocity and the scan number, we found that the best compromise to measure CO$_2$ and CH$_4$ is a scan speed of 120 KHz and a scan number of 100. With this protocol, the total time required to record a spectrum is 83 s, which is low in comparison to the variability of these gases in the atmosphere. Furthermore, a scan velocity of 120 KHz was chosen as a compromise between two important features: the elimination of the ghost signal, which appears at scan velocities below 80 KHz that result from the vibrations of the compressor of

the closed-cycle stirling cooler, and the increase of the detector non-linearity at a velocity of 160 KHz. Note that the spectral parameters are adjustable at each retrieval and for each spectrum in the radiative transfer model ARAHMIS.

4- Although $H_2O$ absorption lines are present in almost all spectral windows, water vapour mole fractions are not retrieved in the analysis. Is it because of the certain meteorological conditions in Izaña that leads to stable water vapour values? Bringing some evidence to prove that's the case would be helpful.

One of the main objectives of the acquisition of CHRIS is the validation of the space instruments, like TANSO-fts/ GOSAT, which has similar spectral bands. Since this information content study has to be easily compared to measurements from space instruments, we considered variability for the water vapor profile as derived from the IASI level 2 products provided by EUMETSAT (Herbin et al. 2013, De Wachter et al. 2017). However, we agree that the impact of water is huge, and in the work that had just begun on the $CO_2$ retrieval, $H_2O$ is part of the retrieved state vector. This information is mentioned in the revised manuscript.

5- In section 3.1, it is mentioned the apriori profiles (I am guessing of $CO_2$ and $CH_4$), temperature and humidity are used for the analysis. Could you please specify where these information are obtained from?

During campaigns, the a priori profiles for the temperature, pressure and water vapor are derived from the radiosondes located both spatially and temporally near our instrument. Otherwise, we take, when available, the radiosondes data located as close as possible to our measurement point. The $CO_2$ and $CH_4$ a priori profiles are interpolated from the available satellite instruments Level 2 profiles and/or NDACC/TCCON a priori profiles.

6- Page 13, the last paragraph, you mention that calculation of $X_G$ using $O_2$ column values done by EM27/SUNs allows comparisons with satellite data and it's not possible for CHRIS. This statement contradicts the point made earlier in the introduction where you suggest CHRIS could be used for satellite validation. In fact, retrieval of $X_G$ values are possible for CHRIS if surface pressure and water vapour measurements are used as described by Wunch et al., 2010.

Indeed, this part of the manuscript might be confusing, and we agree that the retrieval of $X_G$ values is possible for CHRIS if the formula in Wunch et al. 2010 is used. During the MAGIC campaigns, we have access to the balloons and radiosondes data (temperature, surface pressure, $H_2O$ vmr, etc…); so for these particular campaigns, $X_G$ values will be calculated for CHRIS and the results will be compared with the other instruments involved, especially the IFS125HR of the TCCON network and the EM27/SUN, and this will be the subject of the upcoming paper. However, what we want to point out is the fact that the two equations to calculate $X_G$ are not strictly similar since the EM27/SUN can measure $O_2$ column to calculate the DMFs, which eliminates the systematic errors due to the measurements which will not be possible for us, since the $O_2$ band is not detected by CHRIS. This paragraph is rewritten in the revised manuscript to eliminate any ambiguity.